# Dopamine neurons drive fear extinction learning by signaling the omission of expected aversive outcomes

Ximena I Salinas-Hernández[1], Pascal Vogel[1], Sebastian Betz[1], Raffael Kalisch[2,3], Torfi Sigurdsson[1], Sevil Duvarci[1]*

[1]Institute of Neurophysiology, Neuroscience Center, Goethe University, Frankfurt, Germany; [2]Deutsches Resilienz Zentrum, University Medical Center of the Johannes Gutenberg University, Mainz, Germany; [3]Neuroimaging Center, Focus Program Translational Neuroscience, Johannes Gutenberg University, Mainz, Germany

**Abstract** Extinction of fear responses is critical for adaptive behavior and deficits in this form of safety learning are hallmark of anxiety disorders. However, the neuronal mechanisms that initiate extinction learning are largely unknown. Here we show, using single-unit electrophysiology and cell-type specific fiber photometry, that dopamine neurons in the ventral tegmental area (VTA) are activated by the omission of the aversive unconditioned stimulus (US) during fear extinction. This dopamine signal occurred specifically during the beginning of extinction when the US omission is unexpected, and correlated strongly with extinction learning. Furthermore, temporally-specific optogenetic inhibition or excitation of dopamine neurons at the time of the US omission revealed that this dopamine signal is both necessary for, and sufficient to accelerate, normal fear extinction learning. These results identify a prediction error-like neuronal signal that is necessary to initiate fear extinction and reveal a crucial role of DA neurons in this form of safety learning.
DOI: https://doi.org/10.7554/eLife.38818.001

*For correspondence:
duvarci@med.uni-frankfurt.de

**Competing interests:** The authors declare that no competing interests exist.

## Introduction

The ability to learn which stimuli predict danger is crucial for survival but it is equally important to adapt behavior when those stimuli no longer represent a threat. One classic example of this is fear extinction learning, during which the repeated presentation of a stimulus (conditioned stimulus, CS) that no longer predicts an aversive outcome (unconditioned stimulus, US) leads to a gradual decrease in learned fear responses. Many anxiety disorders, such as post-traumatic stress disorder, are characterized by impaired extinction learning (*Craske et al., 2017*; *Graham and Milad, 2011*; *Mahan and Ressler, 2012*; *Milad and Quirk, 2012*; *Pitman et al., 2012*) and thus understanding the neural basis of fear extinction has clinical significance.

A large body of evidence indicates that fear extinction represents new learning rather than forgetting or the erasure of the original fear memory (*Bouton et al., 2006*; *Myers and Davis, 2007*). In order to initiate extinction learning, the absence of the expected aversive outcome must be detected and signaled to the brain regions mediating fear extinction. Decades of research on fear extinction has revealed that a distributed network of brain structures including the amygdala, medial prefrontal cortex and hippocampus mediates the acquisition, consolidation and retrieval of fear extinction memories (*Duvarci and Pare, 2014*; *Maren et al., 2013*; *Pape and Pare, 2010*; *Sotres-Bayon and Quirk, 2010*; *Tovote et al., 2015*). However, none of these structures have been shown to signal the absence of the expected aversive outcome during fear extinction. The neural substrates of such a signal that could initiate extinction learning have therefore remained elusive.

**eLife digest** To survive, animals must identify and react to stimuli in their environment that signal danger. But they must also adapt their behavior when those stimuli no longer signal danger – hiding whenever you hear a loud noise might keep you safe, but it also prevents you from searching for food. In the laboratory, we can study this form of learning using procedures called fear conditioning and extinction.

During fear conditioning, animals learn that a stimulus, such as a tone, signals that an unpleasant event is about to occur. That event might involve receiving a mild shock to the foot, for example. After experiencing the tone and shock paired together multiple times, animals will initially show signs of fear – such as freezing – when they hear the tone. But if later the tone occurs without being followed by the shock, these fear responses fade. This fading process is called extinction. Extinction does not involve erasing the old memory about the tone-shock relationship. That is, it is not a form of forgetting. Instead, the animals learn that the tone no longer signals an impending shock.

By monitoring brain activity in mice trained to associate a shock with a tone, Salinas-Hernández et al. reveal how the brain begins to learn that the shock no longer follows the tone. When the mice do not receive the anticipated shock to the foot, a group of brain cells that produce the chemical dopamine increase their activity. These neurons also fire whenever animals receive a reward, particularly one that exceeds their expectations. The more the dopamine neurons fire, the faster the mice reduce their fear responses to the tone. Preventing the neurons from increasing their activity prevents the mice from extinguishing their fear memory. By contrast, activating the neurons speeds up the extinction process.

Understanding how the brain extinguishes learned fear responses has therapeutic implications. Many anxiety disorders, such as post-traumatic stress disorder, involve impaired fear extinction learning. Indeed, exposure therapy – used to treat anxiety disorders such as phobias – is a form of fear extinction. Manipulating the activity of dopamine neurons during extinction could therefore help to treat anxiety disorders.
DOI: https://doi.org/10.7554/eLife.38818.002

New learning is initiated when outcomes violate expectations (*Rescorla and Wagner, 1972*). Such violations are thought to cause 'prediction error' signals that initiate the neural processes which ultimately lead to changes in behavior (*Friston, 2012*; *den Ouden et al., 2012*). During fear extinction, the absence of the US is an unexpected event and likely generates a prediction error signal that initiates extinction learning. More specifically, the omission of the aversive US can be conceptualized as a better-than-expected outcome. It is well-established that the activity of midbrain dopamine (DA) neurons represents the degree to which outcomes are better or worse than expected (*Bayer and Glimcher, 2005*; *Eshel et al., 2015*; *Eshel et al., 2016*; *Schultz et al., 1997*; *Schultz and Dickinson, 2000*). For example, many DA neurons increase their firing to rewards that are either unexpected or better than expected and this DA signal is sufficient to drive reinforcement learning (*Steinberg et al., 2013*). Based on our data from human studies, we have previously proposed that DA neurons could provide a prediction error-like signal during the aversive US omission to initiate fear extinction (*Raczka et al., 2011*). Consistent with this, an increase in DA release has been observed in the nucleus accumbens (NAc) during fear extinction (*Badrinarayan et al., 2012*) and pharmacological blockade of DA receptors in the NAc impairs fear extinction (*Holtzman-Assif et al., 2010*). However, the electrical activity of DA neurons during fear extinction − particularly at the time of the aversive US omission − and its relationship to extinction learning, is incompletely understood. In this study, we hypothesized that the unexpected omission of the aversive US activates DA neurons in the ventral tegmental area (VTA) and that this signal is necessary to initiate normal fear extinction learning. To test this hypothesis, we used in vivo single-unit recordings, DA neuron-specific calcium recordings and bi-directional optogenetic manipulations in behaving mice to examine and causally test the role of VTA DA neurons in fear extinction.

## Results

### Dopamine neurons signal the unexpected omission of the aversive US during fear extinction learning

We first examined whether DA neurons in the VTA are activated by the unexpected omission of the aversive US during fear extinction. We recorded the single-unit spiking activity of VTA neurons (*Figure 1A–C*;*Figure 1—figure supplement 1*) in mice (n = 11) that were trained in a fear conditioning paradigm (*Figure 1D–F*;*Figure 1—figure supplement 2A*) where a tone (CS) was paired with an aversive foot shock (US). Twenty-four hours after fear conditioning, mice received an extinction session consisting of CS presentations in the absence of the aversive US. A total of 43 (out of 90) and 40 (out of 75) VTA neurons classified as 'putative' DA neurons (see Materials and methods; *Figure 1—figure supplement 3*) were recorded during day 1 and day 2, respectively. Analysis of neuronal firing rates during the time of the US omission revealed that 25% of putative DA neurons (10 of 40) exhibited a significant increase in firing rate to the omission of the aversive US during the early extinction trials (E-Ext: average of first 10 CSs) when the US omission was unexpected (*Figure 1G-I*; *Figure 1—figure supplement 3B*; see Materials and methods for details). This was not simply a response to the CS offset since only 2.3% of neurons (1 of 43; *Figure 1H*;*Figure 1—figure supplement 3A*) showed increased firing at the end of the CS during tone habituation (Hab). On the other hand, during late extinction trials (L-Ext: average of last 10 CSs), when the US omission was no longer unexpected and animals showed significant extinction of fear responses (*Figure 1F*), only 7.5% of putative DA neurons (3 of 40; *Figure 1G-I*) showed an increase in firing to the absence of the US. Importantly, there was no difference in freezing levels during the CS and the post-CS period in E-Ext (*Figure 1—figure supplement 2B*) suggesting that the observed increase in DA neuron firing was not due to an increase in movement when the CS terminates.

Furthermore, analysis of the distribution of z-scores during the US omission (*Figure 2A*) revealed that none of the putative DA neurons (0 of 40) showed a selective decrease in firing to the US omission (see Materials and methods for details) during either E-Ext or L-Ext. This suggests that the dominant response of putative DA neurons to US omission was excitatory. Consistent with this, we observed a significant increase in firing to the US omission at the population level when we examined the average response of all putative DA neurons during E-Ext (paired t-test, t(39) = 2.22, p = 0.03), but not L-Ext (paired t-test, t(39) = 0.20, p = 0.83) or Hab (paired t-test, t(42) = 0.71, p = 0.47; *Figure 1J*).

Local circuit interactions between DA and GABA (γ-aminobutyric acid) neurons in the VTA underlie reinforcement learning (*Cohen et al., 2012*; *Eshel et al., 2015*). It is therefore possible that the DA and local GABA neurons also interact to drive fear extinction learning. For instance, the increased firing of putative DA neurons could be mediated by disinhibition resulting from inhibition of local GABA neurons at the time of the US omission. To test this possibility, we analyzed the activity of putative non-DA neurons – the subset of VTA neurons which likely dominantly includes GABA cells – during the time of the US omission and found that 8.5% of neurons (3 of 35) showed a selective decrease in firing to the US omission during E-Ext (*Figure 1—figure supplement 3B*, *Figure 2B*). However, this was not significantly different from the proportion of non-DA neurons (4.2%, 2 of 47 neurons) showing decreased firing at the CS offset during Hab (*Figure 1—figure supplement 3B*, *Figure 2B*; Fisher's exact test, p = 0.64). These results therefore suggest that the excitation observed in the putative DA neurons during US omission is unlikely mediated by the activity of the local GABA neurons. Moreover, only a small proportion of putative non-DA neurons showed increased firing at the time of the US omission during E-Ext (2.8%, 1 of 35 neurons; *Figure 1—figure supplement 3B*, *Figure 2B*). This proportion was again not different from the 2.1% of neurons (1 of 47) that showed increased firing during Hab (*Figure 1—figure supplement 3B*, *Figure 2B*; Fisher's exact test, p = 1). Together, these results suggest that the putative non-DA, likely local GABA, neurons in VTA do not change their firing at the time of the US omission and that the US omission is signaled specifically by the putative DA neurons in VTA.

In contrast to the uniform response observed during US omission, the responses of putative DA neurons during the CS were diverse. *Figure 1—figure supplement 4* shows example neurons that displayed excitation, inhibition and also biphasic response during the CS. Whereas some neurons showed sustained responses to the CS (*Figure 1—figure supplement 4B*), others showed a transient response at the onset of the CS (*Figure 1—figure supplement 4C*). In order to quantify these

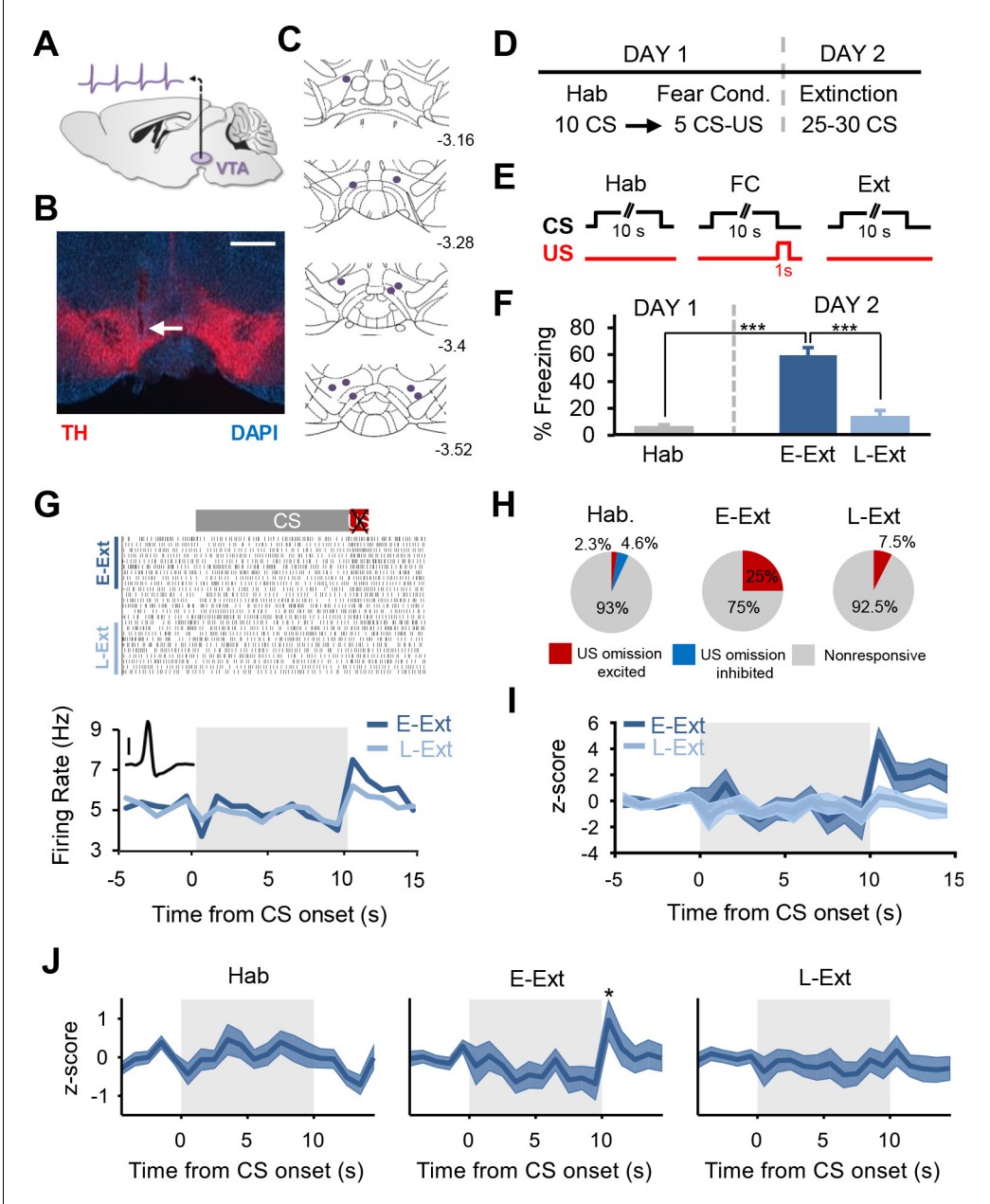

**Figure 1.** Putative dopamine neurons signal the unexpected omission of the aversive US during fear extinction learning. (**A**) Schematic of single-unit recordings in the VTA. (**B**) Histological example showing an electrode placement in the VTA (white arrow). Red: immunostaining against tyrosine hydroxylase (TH), blue: DAPI staining. Scale bar: 0.5 mm. (**C**) Schematic coronal sections showing the location of the recording sites in the VTA. Numbers represent distance posterior to the bregma. (**D**) Schematic of the behavioral protocol. Hab: tone habituation, Fear Cond.: fear conditioning. (**E**) Schematic of CS and US presentations during tone habituation (Hab), fear conditioning (FC) and extinction (Ext). (**F**) Behavioral data. During tone habituation, mice (n = 11) showed low freezing levels in response to the CS. Twenty-four hours after fear conditioning, mice exhibited significant increase in freezing to the CS during E-Ext trials (first 10 CSs, paired t-test comparing E-Ext to Hab, t(10) = 8.30, p<0.0001). The CS evoked low freezing levels during L-Ext trials (last 10 CSs) indicating successful extinction learning (paired t-test: comparing L-Ext to Hab, t(10) = 1.32, p = 0.21; comparing L-Ext to E-Ext, t(10) = 11.9, p<0.0001). (***p<0.0001). (**G**) Raster plot (top) and peristimulus time histogram (1 s bins; bottom) of an example putative DA neuron (see inset; scale bar: 50 μV) responding to US omission during the extinction session. (**H**) The proportion of putative DA neurons that were significantly US omission (or CS offset for Hab) excited, inhibited or nonresponsive during the Hab, E-Ext and L-Ext trials. Proportion of US omission excited neurons significantly increased from Hab

*Figure 1 continued on next page*

*Figure 1 continued*

to E-Ext (Fisher's exact test, p = 0.0028) and decreased back to Hab levels during L-Ext (Fisher's exact test, p = 0.34). Note that there were no US omission inhibited neurons during E-Ext and L-Ext. (I) Peristimulus time histogram showing the z-scored population activity of all putative DA neurons that were significantly US omission excited during E-Ext (n = 10 of 40 putative DA neurons). (J) Peristimulus time histogram showing the z-scored population activity of all putative DA neurons during Hab (left), E-Ext (middle) and L-Ext (right). Note the significant increase in population activity at the time of the US omission during E-Ext (*p<0.05). Shaded regions represent mean ± s.e.m. across neurons.

DOI: https://doi.org/10.7554/eLife.38818.003

The following figure supplements are available for figure 1:

**Figure supplement 1.** Isolation of single-units using steretrodes.
DOI: https://doi.org/10.7554/eLife.38818.004
**Figure supplement 2.** Freezing levels during fear conditioning and the post-CS periods during E-Ext.
DOI: https://doi.org/10.7554/eLife.38818.005
**Figure supplement 3.** Classification of VTA neurons.
DOI: https://doi.org/10.7554/eLife.38818.006
**Figure supplement 4.** CS-evoked responses of putative DA neurons.
DOI: https://doi.org/10.7554/eLife.38818.007

CS-evoked responses, we calculated the mean response to the CS for each neuron during Hab, E-Ext and L-Ext (*Figure 1—figure supplement 4A*). We found that 5% (2 of 40) of putative DA neurons exhibited CS-evoked excitation and 7.5% (3 of 40) CS-evoked inhibition during E-Ext. However these percentages were not significantly different from the proportion of cells that showed excitation (6.9%, 3 of 43 neurons; Fisher's exact test, p = 1) or inhibition (0%, 0 of 43; Fisher's exact test, p = 0.1) during Hab. Furthermore, when we examined the population activity by averaging the response of all putative DA neurons, there was no significant change in the average response during the CS (paired t-tests, Hab: $t(42) = 0.58$, p = 0.56, E-Ext: $t(39) = 1.89$, p = 0.065, L-Ext: $t(39) = 1.23$, p = 0.22; *Figure 1J*;*Figure 1—figure supplement 4A*). These results suggest that the average activity of putative DA neurons in the VTA did not change during the CS even though different subsets of cells showed CS-evoked excitation and inhibition. A lack of strong responses to the CS might be due to the cued fear conditioning paradigm that we used in our study. It has recently been shown that DA neurons respond strongly to the CS when a discriminative fear conditioning task is used and the strength of their response increases with increasing discrimination between the aversive and the safe CS (*Jo et al., 2018*).

The above results show that putative DA, but not non-DA, neurons in the VTA signal the omission of the aversive US during fear extinction, specifically during the beginning of extinction learning when the US omission is unexpected. To further confirm that DA neurons signal the unexpected omission of the US, we next measured activity-dependent calcium signals selectively in DA neurons using fiber photometry. To this end, a Cre-dependent adeno-associated virus (AAV) expressing the genetically encoded calcium ($Ca^{+2}$) indicator gCaMP6 was injected, and an optical fiber implanted, in the VTA of transgenic mice expressing Cre recombinase under the control of the dopamine transporter (Dat) promoter (DAT-Cre mice; *Figure 3A–B* and *Figure 3—figure supplement 1A*). In these mice, Cre expression is highly selective for dopamine neurons (*Lammel et al., 2015*). Accordingly, we observed a high degree of overlap between Cre-dependent gCaMP6 expression and immunohistochemical staining against tyrosine hydroxylase (TH; *Figure 3—figure supplement 1B–C*). In control mice, we injected a Cre-dependent AAV expressing GFP to examine whether changes in fluorescence could be independent of neuronal activity. Recordings from gCaMP6-expressing animals revealed transient increases in fluorescence whereas such increases were absent in mice expressing GFP (*Figure 3C*). Furthermore, consistent with electrophysiological studies in VTA DA neurons (*Eshel et al., 2015*; *Eshel et al., 2016*; *Roesch et al., 2007*), we also confirmed that reward delivery caused large increases in fluorescence in the gCaMP6-expressing mice (*Figure 3—figure supplement 2*, also see Materials and methods), indicating that this $Ca^{+2}$ signal is indeed generated by DA neuron activity.

Both gCaMP6- and GFP-expressing animals underwent the same fear conditioning protocol as in the electrophysiology experiment (*Figure 3D–E*) and showed comparable levels of freezing to the

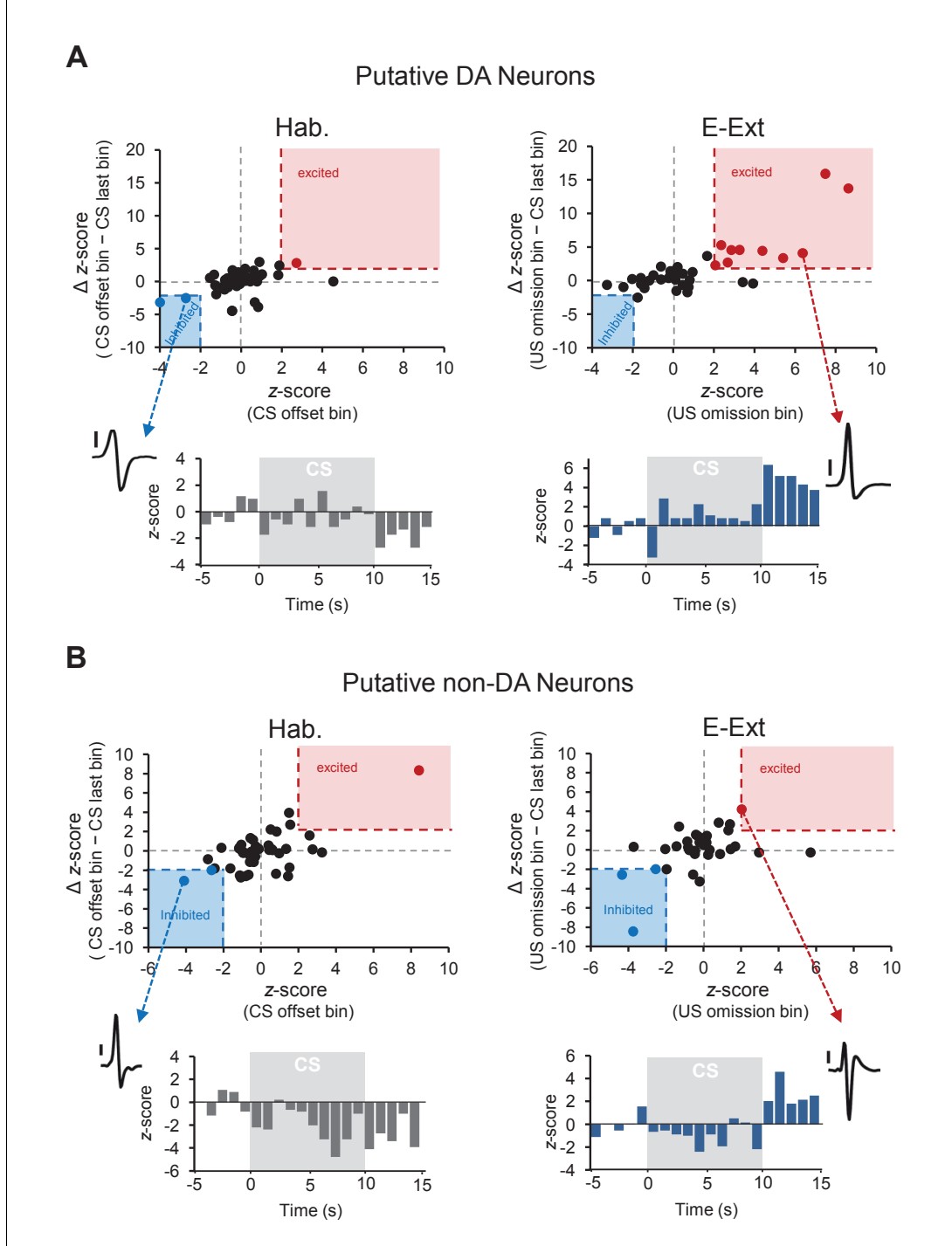

**Figure 2.** Distribution of z-scores during the US omission for putative DA and non-DA neurons. (**A**) Top: plot of z-score (CS offset bin) against the Δ z-score (CS offset bin − CS last bin) during tone habituation (Hab.; left) and z-score (US omission bin) against the Δ z-score (US omission bin − CS last bin) during early extinction (E-Ext; right). Each dot represents a putative DA neuron. The red shaded areas (x-axis: 2, y-axis: 2) contain the significantly excited (red dots) and the blue shaded areas (x-axis: −2, y-axis: −2) contain significantly inhibited (blue dots) neurons. Black dots represent non-responsive neurons. Bottom: peristimulus time histogram (1 s bins) of an example neuron (see inset; scale bar: 50 µV) that showed significant inhibition to CS offset during Hab (left) and of an example neuron (see inset; scale bar: 50 µV) that showed significant excitation to the US omission during E-Ext (right). (**B**) The same as in (**A**), but for putative non-DA neurons.

DOI: https://doi.org/10.7554/eLife.38818.008

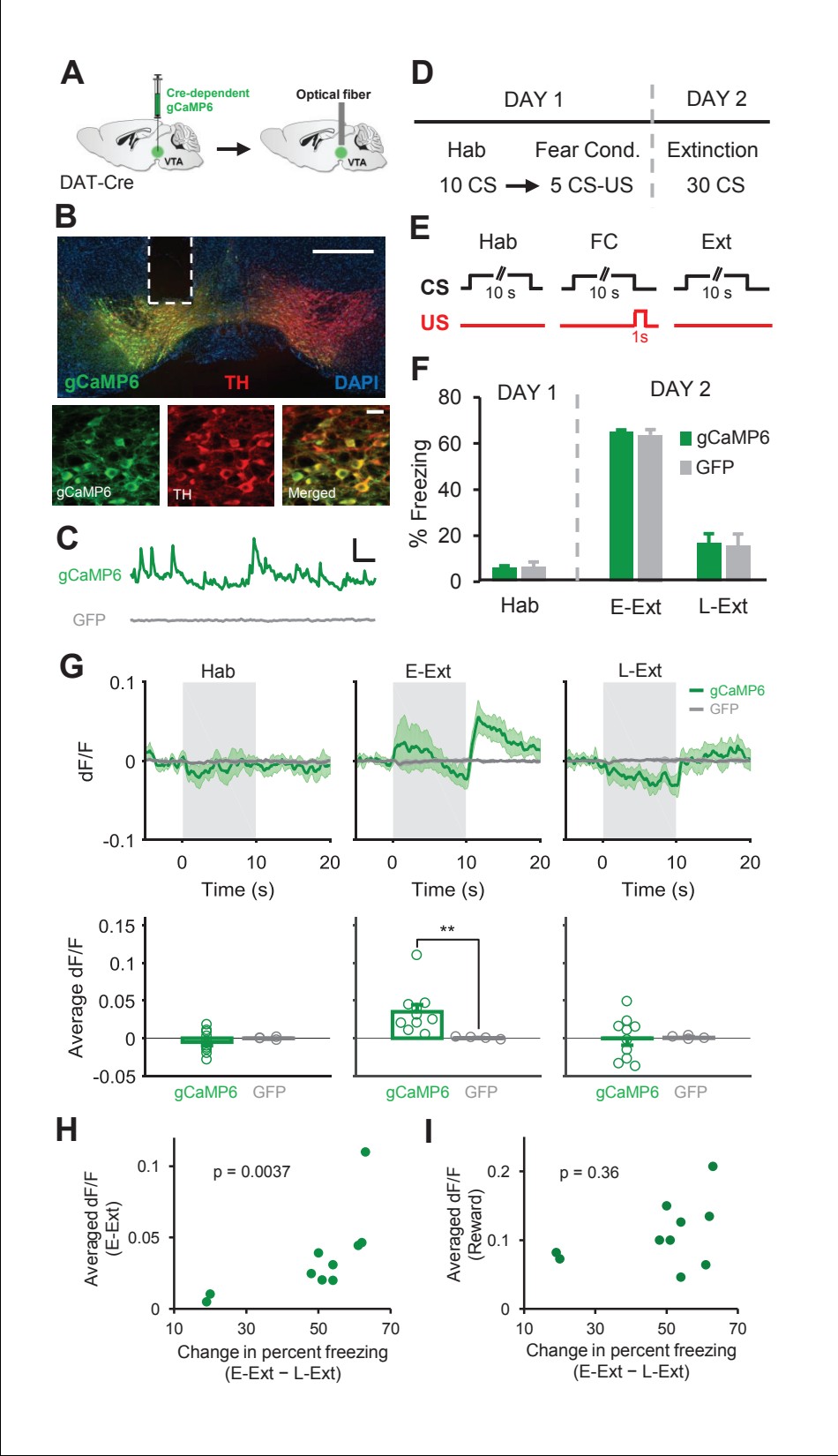

**Figure 3.** Calcium recordings in VTA dopamine neurons confirm signaling of the unexpected US omission during fear extinction learning. (**A**) Schematic of the surgical procedure showing the virus injection (left) and optical fiber

*Figure 3 continued on next page*

*Figure 3 continued*

implantation (right) in the VTA. (**B**) Top: example histological image showing Cre-dependent expression of gCaMP6 (green) along with immunostaining for tyrosine hydroxylase (TH, red) and DAPI (blue) staining in the VTA. White vertical track indicates the optical fiber placement in the VTA. Scale bar: 0.5 mm. Bottom: confocal images showing expression of gCaMP6 (left), TH (middle) and a merged image (right) showing co-expression. Scale bar: 20 μm. (**C**) Examples of changes in fluorescence (dF/F) over time in an animal expressing gCaMP6 (green) and an animal expressing the control fluorophore GFP (gray). Scale bar: 5 s, 0.2 dF/F. (**D**) Schematic of the behavioral protocol. Hab: tone habituation, Fear Cond.: fear conditioning. (**E**) Schematic of CS and US presentations during tone habituation (Hab), fear conditioning (FC) and extinction (Ext). (**F**) Behavioral freezing to the CS during tone habituation and extinction session for gCaMP6 (n = 10) and GFP (n = 4) groups. During Hab, all mice showed low freezing levels in response to the CS. Twenty-four hours after fear conditioning, both groups of mice increased freezing to the CS during E-Ext (first 10 CSs). The CS evoked low freezing levels during L-Ext (last 10 CSs) indicating successful extinction learning. (**G**) Top: Average change in fluorescence in animals expressing gCaMP6 (green, n = 10) or GFP (gray, n = 4) around the time of CS presentation (gray area) during Hab, E-Ext and L-Ext. Note the increase in fluorescence at the offset of the CS (the time of the unexpected US omission) during E-Ext. Bottom: Average change in fluorescence in the 5 s after CS offset during Hab, E-Ext and L-Ext. gCaMP6-expressing animals exhibited a significant increase in $Ca^{+2}$ signal during E-Ext compared to the GFP control group (\*\*p<0.01, rank-sum test). (**H**) Correlation between the average change in fluorescence in the 5 s after CS offset during E-Ext and the change in percent freezing from E-Ext to L-Ext (n = 10 mice; Spearman's correlation = 0.83, p = 0.0037). (**I**) Correlation between the average change in fluorescence during the 0–3 s after reward delivery and change in percent freezing from E-Ext to L-Ext (n = 10 mice; Spearman's correlation = 0.32, p = 0.36). Shaded regions and error bars represent mean ±s.e.m across animals.

DOI: https://doi.org/10.7554/eLife.38818.009

The following figure supplements are available for figure 3:

**Figure supplement 1.** Placement of optical fibers and DA neuron-specific expression of GCaMP6.
DOI: https://doi.org/10.7554/eLife.38818.010
**Figure supplement 2.** Responses of VTA DA neurons to reward.
DOI: https://doi.org/10.7554/eLife.38818.011
**Figure supplement 3.** Freezing levels during fear conditioning and the post-CS periods during E-Ext.
DOI: https://doi.org/10.7554/eLife.38818.012

CS across sessions (two-way repeated measures ANOVA; main effect of group: $F_{1,24}$ = 0.04, p = 0.83; group × trial interaction: $F_{2,24}$ = 0.04, p = 0.96; *Figure 3F* and *Figure 3—figure supplement 3A*). During E-Ext, we observed a significant increase in the $Ca^{+2}$ signal of gCaMP6 animals at the time of the US omission compared to the pre-CS baseline (p<0.01, sign-rank test) and the GFP control group (p<0.01, rank-sum test; *Figure 3G*). On the other hand, during L-Ext, when the US omission was no longer unexpected, the $Ca^{+2}$ signal did not change during the post-CS period (p = 0.85, sign-rank test; *Figure 3G*). No changes in fluorescence were observed in GFP-expressing animals during either E-Ext or L-Ext (*Figure 3G*). Furthermore, we did not observe any change in the $Ca^{+2}$ signal during Hab (gCaMP6 group, p = 0.32, sign-rank test; *Figure 3G*) suggesting that the increase during US omission is unlikely a response to the CS offset. Furthermore, freezing levels during the CS and the post-CS period in E-Ext were comparable (*Figure 3—figure supplement 3B*) ruling out the possibility that an increase in movement when CS terminates might have resulted in the observed increase in $Ca^{+2}$ signal at the time of the US omission during E-Ext. Notably, these results obtained by measuring the population $Ca^{+2}$ signal from DA neurons are consistent with the results of the electrophysiology recordings where we found a significant increase in the average population activity of all putative DA neurons at the time of the US omission during E-Ext (*Figure 1J*).

If activation of DA neurons at the time of the unexpected US omission drives extinction learning then this $Ca^{+2}$ signal during E-Ext should be larger in animals that exhibit better extinction learning. To test this, we took advantage of the variability in the extinction learning rates of individual mice and asked whether they were correlated with the $Ca^{+2}$ signal at the time of the US omission during E-Ext. This revealed a significant correlation between the $Ca^{+2}$ signal during E-Ext and the change in freezing from E-Ext to L-Ext (Spearman's correlation = = 0.83, p = 0.0037; *Figure 3H*). A significant correlation was also observed between the $Ca^{+2}$ signal during E-Ext and the freezing levels during L-Ext (Spearman's correlation = 0.92, p = 0.0003). However, it is possible that the variation in the $Ca^{+2}$ signal during E-Ext might be due to the variation in recording locations in the VTA across

animals rather than reflecting the relationship with extinction learning. We reasoned that differences in recording locations would likely result in variation in reward responses. We therefore examined the correlation between the change in freezing from E-Ext to L-Ext and reward responses and did not find a significant relationship between these two variables (Spearman's correlation = 0.32, p = 0.36; *Figure 3I*). These results therefore suggest that the magnitude of the $Ca^{+2}$ signal during E-Ext correlated with the level of extinction learning and the variations in the magnitude of this signal were not due to differences in the recording location.

Contrasting with the uniform increase in DA neuron activity during US omission (*Figure 3G*), the responses to the CS varied across animals (*Figure 4A*). Some animals showed increased and some decreased fluorescence during the CS in E-Ext (*Figure 4A–B*). Accordingly, consistent with single unit results which showed no change in CS-evoked population activity of putative DA neurons

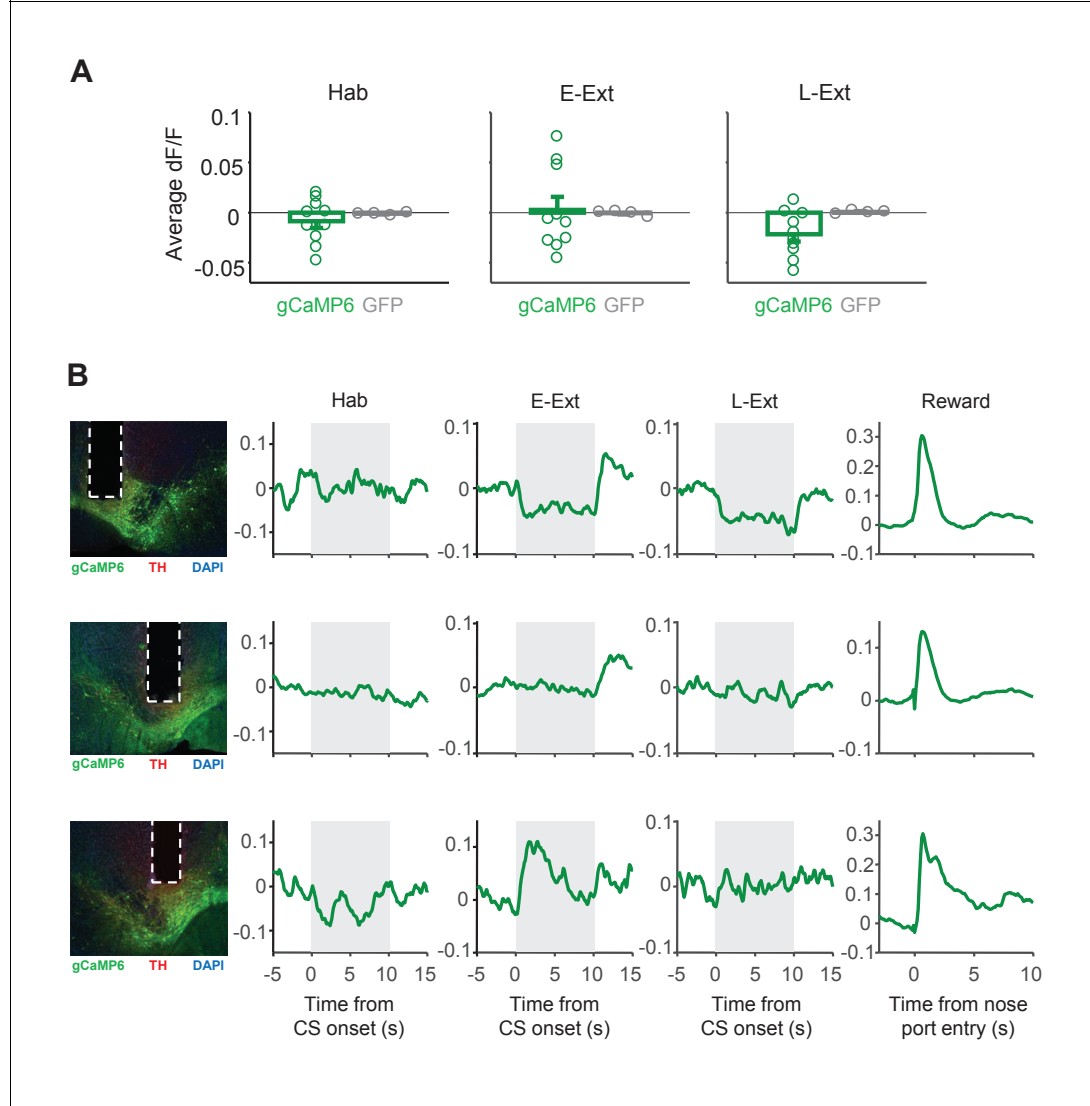

**Figure 4.** Changes in the calcium signal during CS presentations. (**A**) Average change in fluorescence (dF/F) during the CS (10 s) during tone habituation (Hab), early extinction (E-Ext) and late extinction (L-Ext). There was no difference in responses to the CS between the gCaMP6-expressing (n = 10) and GFP-expressing (n = 4) animals (rank-sum test: Hab, p = 1.0; E-Ext, p = 0.37; L-Ext, p = 0.11). (**B**) Example CS responses in gCaMP6-expressing animals that showed decreased (top), unchanged (middle) and increased fluorescence during the CS (gray area). Left: histology image showing the location of the optical fiber in VTA. Green: cre-dependent gCaMP6 expression, red: immunostaining for tyrosine hydroxylase (TH), blue: DAPI staining. White vertical tracks indicate the optical fiber placement in the VTA. Right: Changes in fluorescence during Hab, E-Ext, L-Ext and Reward.

DOI: https://doi.org/10.7554/eLife.38818.013

(*Figure 1J*; *Figure 1—figure supplement 4A*), the $Ca^{+2}$ signal evoked by the CS was not significantly different from the baseline during E-Ext (p = 1.0, sign-rank test) or Hab (p = 0.43, sign-rank test) when we averaged the CS responses of all animals (*Figure 4A*).

Overall, these findings together with our single-unit results demonstrate that DA neurons signal the unexpected omission of the aversive US during fear extinction and that the magnitude of this DA signal predicts the strength of extinction learning.

## Inhibition of dopamine neuron firing at the time of the US omission impairs fear extinction learning

We next asked whether the observed increase in DA neuron firing at the time of the unexpected US omission is necessary for fear extinction learning. To address this question, we optogenetically inhibited DA neurons in the VTA at the time of the US omission during fear extinction. DAT-cre mice received bilateral injections of a Cre-dependent AAV expressing either the light-activated inhibitory opsin halorhodopsin fused with enhanced yellow fluorescent protein (NpHR-eYFP) or eYFP only (eYFP control) into the VTA, as well as bilateral implantation of optical fibers above the VTA to allow for selective inhibition of VTA DA neurons (*Figure 5A–C* and *Figure 5—figure supplement 1A*). We observed a high degree of overlap between Cre-dependent NpHR-eYFP expression and immunohistochemical staining against TH (*Figure 5—figure supplement 1B–C*) suggesting DA neuron-selective expression. Furthermore, we confirmed that optical stimulation of NpHR inhibits DA neuron firing in awake DAT-cre mice (*Figure 5—figure supplement 2*).

Mice were trained in a fear conditioning protocol (*Figure 5D*) consisting of 4 CS-US pairings on day 1. Twenty-four hours after fear conditioning, mice received an extinction session. In the experimental group expressing NpHR-eYFP light was delivered bilaterally to the VTA to inhibit DA neurons specifically at the end of each CS presentation, that is during the time of the US omission (Paired-NpHR, n = 7; *Figure 5E*). The behavior of the experimental group was compared to two control groups: one group consisted of mice expressing eYFP only which received the identical light delivery (Paired-eYFP, n = 7) and a second group consisted of mice expressing NpHR-eYFP that received light delivery to inhibit DA neurons during the intertrial intervals (ITIs; Unpaired-NpHR, n = 8; *Figure 5F*).

Compared to the two control groups, the Paired-NpHR group exhibited high freezing levels to the CS throughout the extinction session, suggesting impaired extinction learning (*Figure 5G*). A two-way repeated measures ANOVA revealed a significant main effect of group ($F_{2,456}$ = 7.03, p = 0.0052) and a significant interaction between group and trial ($F_{48,456}$ = 2.19, p<0.0001). Consistent with this, there was a significant difference between the Paired-NpHR group and the Paired-eYFP (p<0.001) or Unpaired-NpHR (p<0.001) controls during L-Ext (*Figure 5I*). Furthermore, impaired extinction learning resulted in a weaker memory for extinction when tested the next day (extinction recall test; two-way repeated measures ANOVA; main effect of group: $F_{2,456}$ = 7.1, p = 0.005; group × trial interaction: $F_{48,456}$ = 1.49, p = 0.02; *Figure 5G*). Consistently, during the early extinction recall trials (E-Ext Rec: first 10 CSs) the Paired-NpHR group froze significantly more compared to the Paired-eYFP (p<0.001) or the Unpaired-NpHR (p<0.001) controls (*Figure 5J*). In contrast to the Paired-NpHR group, the Unpaired-NpHR group behaved comparable to the Paired-eYFP control group during both extinction and extinction recall tests (*Figure 5G–J*) suggesting that optical inhibition of DA neurons per se did not result in a nonspecific increase in freezing levels, and that the behavioral effect was dependent on the temporally specific inhibition of DA neurons during the time of the US omission. Furthermore, there was no difference between the groups in their fear acquisition on day 1 (two-way repeated measures ANOVA, no main effect of group, $F_{2,57}$ = 0.3, p = 0.74 or no group × trial interaction, $F_{6,57}$ = 0.94, p = 0.47; *Figure 5G*) and all groups showed comparable levels of freezing at the start of extinction (first CS; one-way ANOVA, $F_{2,19}$ = 2.06, p = 0.15; *Figure 5H*) before any experimental manipulation took place, ruling out the possibility that differences in the strength of fear memory on day 2 between groups might have caused the observed effect.

Importantly, freezing levels of the NpHR group at the beginning of extinction recall were comparable to freezing levels at the beginning of extinction (paired t-test comparing first CS of extinction and first CS of extinction recall, t(6) = 0.56, p = 0.59), suggesting that no significant extinction learning happened in these animals. Furthermore, we found that the extinction rate of the Paired-NpHR group during extinction recall, in the absence of optogenetic inhibition, was comparable to the

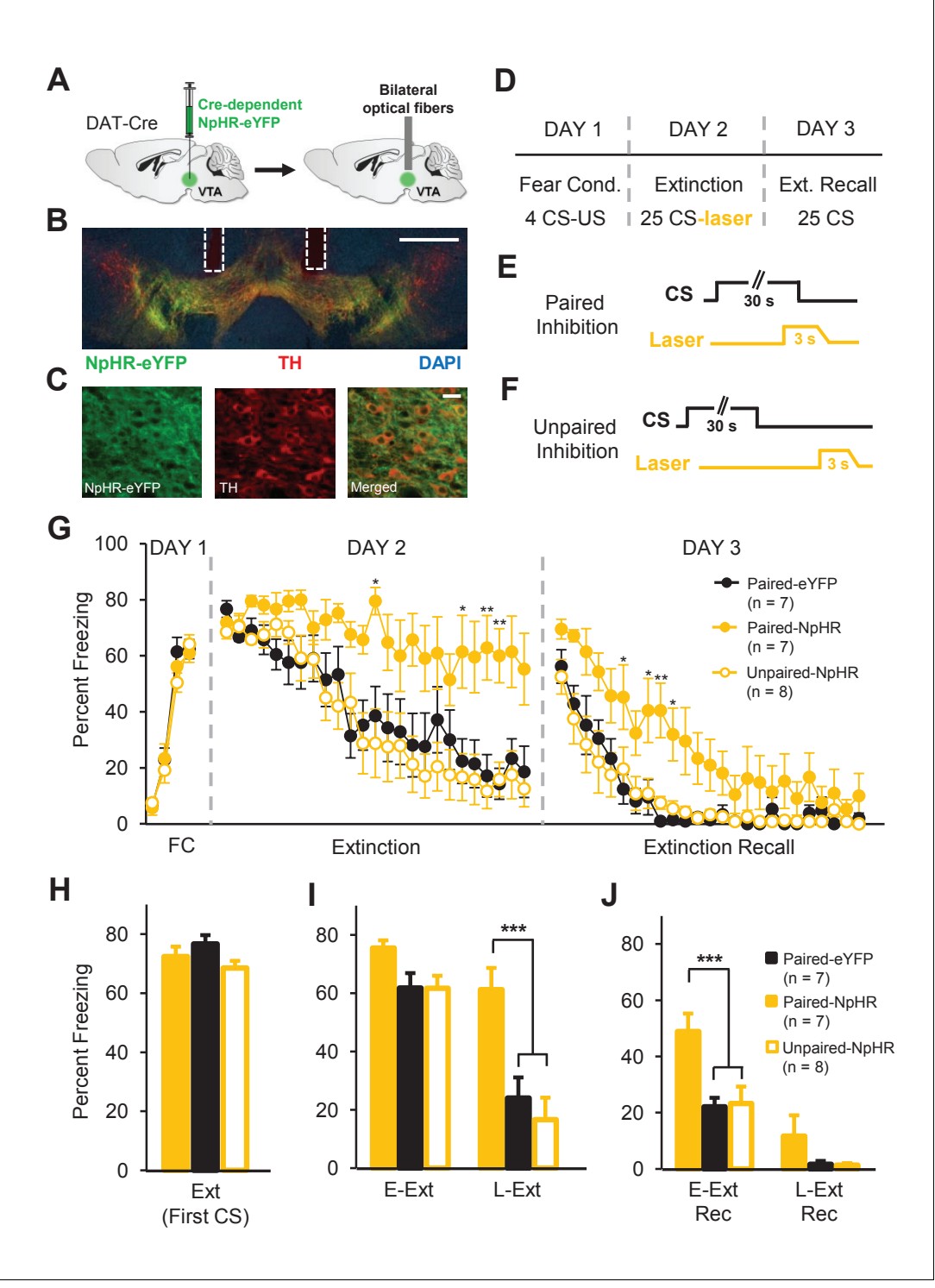

**Figure 5.** Inhibition of dopamine neuron firing at the time of the US omission impairs fear extinction learning. (A) Schematic of the surgical procedure showing bilateral virus injection (left) and optical fiber implantation (right) in the VTA. (B) Example histological image showing Cre-dependent expression of NpHR-eYFP (green) along with immunostaining for tyrosine hydroxylase TH (red) and DAPI (blue) staining in the VTA. White vertical tracks indicate the bilateral optical fiber placements in the VTA. Scale bar: 0.5 mm. (C) Confocal images showing expression of NpHR-eYFP (left), TH (middle) and merged image (right) showing co-expression. Scale bar: 20 μm. (D) Schematic of the behavioral protocol. Fear Cond.: fear conditioning, Ext Recall: extinction recall. (E) Schematic of paired optogenetic inhibition of DA neurons at the time of the US omission. (F) Schematic of unpaired

*Figure 5 continued on next page*

*Figure 5 continued*
optogenetic inhibition during intertrial intervals. (**G**) Percent freezing to the CS during fear conditioning (FC), extinction and extinction recall sessions. The Paired-NpHR group showed impaired extinction learning and extinction recall. (\*\*p<0.01, \*p<0.05). (**H**) No difference in freezing to the CS between groups at the start of extinction (first CS). Ext: extinction. (**I**) Freezing levels during E-Ext (average of first 10 CSs) and L-Ext (average of last 10 CSs; two-way repeated measures ANOVA, main effect of group: $F_{2,19}$ = 9.05, p = 0.0017; group × trial interaction: $F_{2,19}$ = 7.38, p = 0.0043). The Paired-NpHR group (n = 7) showed significantly higher freezing to the CS compared to the Paired-eYFP (n = 7) and Unpaired-NpHR (n = 8) groups during L-Ext trials (\*\*\*p<0.001). E-Ext: early extinction, L-Ext: late extinction. (**J**) Freezing levels during E-Ext Rec (average of first 10 CSs) and L-Ext Rec trials (average of last 10 CSs; two-way repeated measures ANOVA, main effect of group: $F_{2,19}$ = 7.21, p = 0.0047). The Paired-NpHR group exhibited significantly higher freezing to the CS compared to the control groups during E-Ext Rec (\*\*\*p<0.001). E-Ext Rec: early extinction recall, L-Ext Rec: late extinction recall. Error bars represent mean ± s.e.m. across animals.

DOI: https://doi.org/10.7554/eLife.38818.014
The following figure supplements are available for figure 5:

**Figure supplement 1.** Placement of optical fibers and DA neuron-specific expression of NpHR-eYFP.
DOI: https://doi.org/10.7554/eLife.38818.015
**Figure supplement 2.** Optical activation of NpHR inhibits DA neuron firing in awake behaving mice.
DOI: https://doi.org/10.7554/eLife.38818.016

extinction rate of the Paired-eYFP and Unpaired-NpHR groups during extinction (two-way repeated measures ANOVA; no main effect of group $F_{2,456}$ = 1.02, p = 0.37 and group × trial interaction: $F_{48,456}$ = 0.86, p = 0.73). This suggests that our manipulation did not have a nonspecific long-term effect on the ability of the Paired-NpHR group to exhibit extinction learning. These results also suggest that our optogenetic manipulation likely did not affect the strength of the fear memory. Taken together, these findings demonstrate that DA neuron activation by the unexpected omission of the US is necessary for fear extinction learning.

## Enhancing dopamine neuron firing at the time of the US omission accelerates fear extinction learning

If DA neuron firing at the time of the unexpected US omission drives fear extinction, then enhancing this DA signal should accelerate fear extinction learning. To test this, we optogenetically excited DA neurons precisely at the time of the US omission during fear extinction learning. DAT-cre mice were bilaterally injected with a Cre-dependent AAV expressing either channelrhodopsin-2 (ChR2) fused with eYFP (ChR2-eYFP) or eYFP only (eYFP control) into the VTA, and implanted bilaterally with optical fibers above VTA (*Figure 6A–C*; *Figure 6—figure supplement 1A*). There was again a high level of overlap between Cre-dependent ChR2-eYFP expression and immunohistochemical staining against TH (*Figure 6—figure supplement 1B–C*) suggesting DA neuron-selective expression of ChR2. Furthermore, we confirmed that optical stimulation of ChR2 induces firing of DA neurons in awake DAT-cre mice (*Figure 6—figure supplement 2*).

Mice were trained in a fear conditioning protocol (*Figure 6D*) similar to the optogenetic inhibition experiment. The experimental group consisted of ChR2-eYFP expressing mice which received light stimulation of DA neurons specifically at the time of the US omission (Paired-ChR2, n = 7; *Figure 6E*). Two control groups, one expressing eYFP only that received the identical light delivery (Paired-eYFP, n = 7) and the other expressing ChR2-eYFP that received light excitation during the ITIs (Unpaired-ChR2, n = 7; *Figure 6F*), were used to control for nonspecific effects of light and DA neuron stimulation, respectively.

As expected, all groups showed a gradual decrease in freezing to the CS during the extinction session. However, in the Paired-ChR2 group, freezing decreased faster than in the control groups suggesting accelerated extinction learning (*Figure 6G*). A two-way repeated measures ANOVA comparing freezing levels confirmed this observation by revealing a significant main effect of group ($F_{2,432}$ = 4.1, p = 0.03). Comparison of freezing levels in the three groups, particularly during E-Ext, revealed a significant difference between the Paired-ChR2 group and the Paired-eYFP (p<0.05) or Unpaired-ChR2 (p<0.05) controls (*Figure 6I*) suggesting accelerated extinction learning. On the other hand, the two control groups behaved comparably (p>0.05). These results suggest that optical

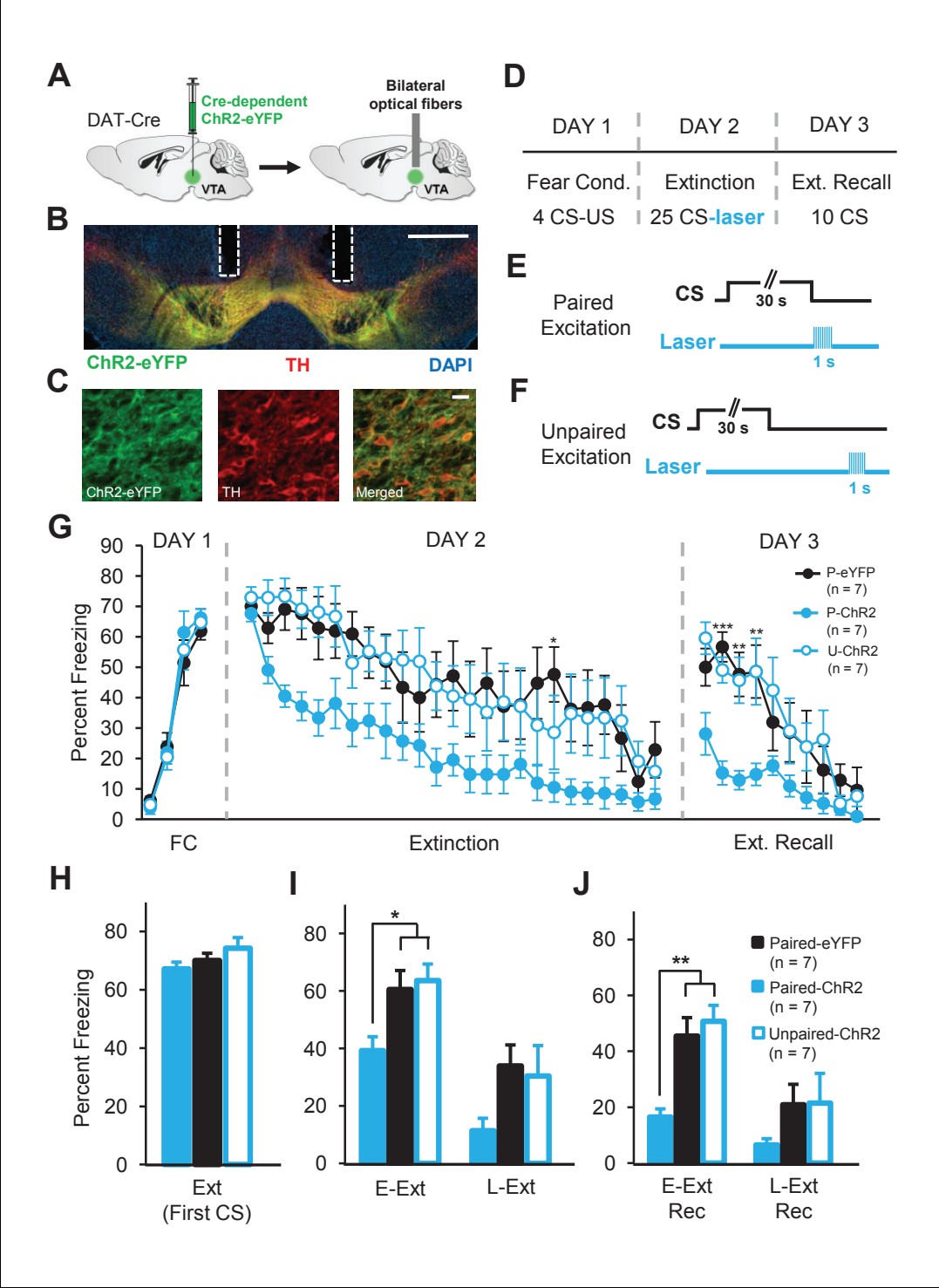

**Figure 6.** Enhancing dopamine neuron firing at the time of the US omission accelerates fear extinction learning. (**A**) Schematic of the surgical procedure showing the bilateral virus injection (left) and optical fiber implantation (right) in the VTA. (**B**) Example histology showing Cre-dependent expression of ChR2-eYFP (green) along with immunostaining for tyrosine hydroxylase (TH, red) and DAPI (blue) staining in the VTA. White vertical tracks indicate the bilateral optical fiber placements in the VTA. Scale bar: 0.5 mm. (**C**) Confocal images showing expression of ChR2-eYFP (left), TH (middle) and merged image (right) showing co-expression. Scale bar: 20 μm. (**D**) Schematic of the behavioral protocol. Fear Cond.: fear conditioning, Ext Recall: extinction recall. (**E**) Schematic of the paired optogenetic excitation of DA neurons at the time of the US omission. (**F**) Schematic of the unpaired

*Figure 6 continued on next page*

*Figure 6 continued*

optogenetic excitation during intertrial intervals. (**G**) Percent freezing to the CS during fear conditioning (FC), extinction and extinction recall sessions. The Paired-ChR2 (P-ChR2) group showed accelerated extinction learning and better extinction recall. (\*\*\*p<0.001, \*\*p<0.01, \*p<0.05). (**H**) No difference in freezing to the CS between groups at the start of extinction (first CS). Ext: extinction. (**I**) Freezing levels during E-Ext (average of first 10 CSs) and L-Ext (average of last 10 CSs; two-way repeated measures ANOVA, significant group effect: $F_{2,18} = 3.88$, p = 0.03). The P-ChR2 group (n = 7) exhibited significantly lower freezing compared to the Paired-eYFP (P-eYFP; n = 7) and Unpaired-ChR2 (U-ChR2; n = 7) control groups during E-Ext (\*p<0.05). E-Ext: early extinction, L-Ext: late extinction. (**J**) Freezing levels during E-Ext Rec (average of first 5 CSs) and L-Ext Rec (average of last 5 CSs; two-way repeated measures ANOVA, main effect of group: $F_{2,18} = 5.5$, p = 0.01 and group × trial interaction: $F_{2,18} = 4.81$, p = 0.021). The P-ChR2 group exhibited significantly lower freezing to the CS compared to control groups during E-Ext Rec (\*\*p<0.01). E-Ext Rec: early extinction recall, L-Ext Rec: late extinction recall. Data are presented as means and error bars represent s.e.m.

DOI: https://doi.org/10.7554/eLife.38818.017

The following figure supplements are available for figure 6:

**Figure supplement 1.** Placement of optical fibers and DA neuron specific expression of ChR2-eYFP.
DOI: https://doi.org/10.7554/eLife.38818.018

**Figure supplement 2.** Optical stimulation of ChR2 induces firing of DA neurons in awake behaving mice.
DOI: https://doi.org/10.7554/eLife.38818.019

**Figure supplement 3.** All animals showed comparable levels of fear renewal.
DOI: https://doi.org/10.7554/eLife.38818.020

excitation of DA neurons per se did not result in a nonspecific decrease in freezing levels and that the temporally precise excitation during the US omission is necessary for the observed behavioral effect. Furthermore, there was no difference between the groups in their fear acquisition on day 1 (two-way repeated measures ANOVA, no main effect of group, $F_{2,54} = 0.17$, p = 0.84 and no group × trial interaction, $F_{6,54} = 0.65$, p = 0.68, *Figure 6G*) and all groups showed comparable levels of freezing during the first CS of extinction (one-way ANOVA, $F_{2,18} = 0.81$, p = 0.66, *Figure 6H*) before light stimulation began. Thus, excitation of DA neurons precisely at the time of the unexpected US omission is sufficient to accelerate fear extinction learning. Finally, Paired-ChR2 mice spent less time freezing to the CS compared to the control groups during the extinction recall test (two-way repeated measures ANOVA; main effect of group: $F_{2,162} = 5.5$, p = 0.013 and group × trial interaction: $F_{18,162} = 2.52$, p = 0.0011, *Figure 6G*) suggesting that the accelerated extinction learning resulted in a stronger extinction memory.

An alternative possibility is that the low level of freezing in the Paired-ChR2 group during Ext Rec was due to an effect of optical stimulation of DA neurons on the fear memory. For instance, the optogenetic manipulation could result in the erasure of the fear memory by impairing the memory reconsolidation process (*Nader, 2015*) rather than accelerating extinction learning and strengthening extinction memory. To rule this possibility out, we tested the animals on a fear renewal test on day 4 by presenting 5 CSs in the conditioning context. It is well established that extinction learning is context-dependent such that if the animals are tested in a different context than the one they are extinguished in, fear responses return, a phenomenon called fear renewal (*Bouton, 2004*). Therefore, if low level of freezing during Ext Rec was due to impaired reconsolidation then we would expect to see impaired fear renewal in Paired-ChR2 group (*Duvarci and Nader, 2004*). However, we found that all groups showed high freezing to the CS during the renewal test and that there was no difference in the freezing levels between the groups (*Figure 6—figure supplement 3*; two-way repeated measures ANOVA, no significant effect of group, $F_{2,72} = 0.33$, p = 0.72 and group × trial interaction, $F_{8,72} = 0.86$, p = 0.55) suggesting that all three groups showed comparable levels of fear renewal. This suggests that the low level of freezing in Paired-ChR2 group during extinction recall was not due to an affect of our optogenetic manipulation on the fear memory but rather was due to enhanced extinction learning and memory formation. Taken together, these findings demonstrate that increasing DA neuron activity at the time of US omission — and thus enhancing an endogenous extinction mechanism — is sufficient to accelerate extinction learning and strengthen extinction memory.

## Discussion

Here we demonstrated that DA neurons were activated by the omission of the aversive US during fear extinction, specifically during the beginning of extinction when the US omission is most unexpected. Importantly, the magnitude of this DA signal correlated with the strength of extinction learning. Furthermore, temporally specific optogenetic inhibition of DA neurons at the time of the US omission prevented extinction, demonstrating that this signal is necessary for normal fear extinction. Conversely, enhancing this DA signal using temporally-specific optogenetic excitation was sufficient to accelerate extinction learning. Together, these results identify a crucial role of DA neurons in signaling the unexpected omission of aversive outcomes and thereby driving fear extinction learning.

Previous studies have shown that DA neurons encode a reward prediction error, or the discrepancy between expected and actual rewards, which acts as a teaching signal for reinforcement learning (*Bayer and Glimcher, 2005*; *Eshel et al., 2015*; *Eshel et al., 2016*; *Schultz et al., 1997*; *Steinberg et al., 2013*). Specifically, presentation of unexpected or better than expected rewards induces increased firing in DA neurons (*Bayer and Glimcher, 2005*; *Eshel et al., 2016*; *Roesch et al., 2007*; *Schultz et al., 1997*). Our results suggest that DA neurons might also signal a better than expected outcome during fear extinction. We found that a subpopulation of putative DA neurons in the VTA increased their firing selectively at the time of the US omission during fear extinction. This US omission-responsive firing was observed specifically during the early trials of extinction when the absence of the US was unexpected, and was significantly reduced during the late stages of extinction when the US omission was no longer unexpected and animals showed significant extinction of fear responses. Importantly, these responses were not observed in putative non-DA neurons. This suggests that DA neurons encode a prediction error-like signal during fear extinction learning. $Ca^{+2}$ recordings selectively in DA neurons further confirmed these results and revealed that this DA signal correlated with the strength of extinction learning. Recent studies have shown that DA neurons not only encode reward prediction errors but also signal prediction errors to gate fear learning (*Groessl et al., 2018*) and to drive threat avoidance (*Menegas et al., 2018*). Interestingly, omission of aversive stimuli during fear extinction in fruit flies is encoded by the DA system that mediates reward, but not aversive, learning (*Felsenberg et al., 2018*). Whether the DA signal during fear extinction in mammals is also similar to prediction error signals for reward and mediated by the brain's reward circuitry (*Wise, 2002*) will be important questions for future studies.

Consistent with our results, previous studies using partial reinforcement paradigms have shown that DA neurons exhibit increased firing to the unexpected omission of aversive stimuli (*Matsumoto and Hikosaka, 2009*; *Matsumoto et al., 2016*; but see *Tian and Uchida, 2015*). However, responses to aversive US omission in these studies were much smaller compared to what we observed. There are several differences between these studies and ours that might account for this. The aversive US used in our study is a painful footshock whereas an air-puff was used in previous studies (*Matsumoto and Hikosaka, 2009*; *Matsumoto et al., 2016*). Furthermore, it has been shown that the valence of the testing context influences DA neuron responses to the omission of the aversive US. Matsumoto and colleagues (2016) found a significant increase in DA neuron responses to the omission of aversive airpuff in a low reward, but not high reward, context. In our study, the animals were only fear conditioned and no reward learning happened prior to or during fear conditioning. Therefore, the context was exclusively aversive. In addition, the CS-US contingency during fear conditioning was higher in our study than in partial reinforcement tasks used in previous studies. The CS predicted the shock with 100% probability at the end of fear conditioning and therefore, the omission of the US was fully unexpected at the beginning of fear extinction in our study. On the other hand, in the studies using partial reinforcement tasks the CS-US contingency was 25–90% and the omission of the US was performed intermittently (*Matsumoto and Hikosaka, 2009*; *Matsumoto et al., 2016*; *Tian and Uchida, 2015*). The US omission was therefore arguably less unexpected. Together, these factors might therefore account for the differences in DA neuron responses between these studies and ours. Overall, our findings suggest that DA neurons not only signal better than expected rewards (*Bayer and Glimcher, 2005*; *Eshel et al., 2016*; *Roesch et al., 2007*; *Schultz et al., 1997*) but also better than expected outcomes more generally, such as the omission of an aversive event.

Detecting the discrepancy between expected and actual outcomes is critical for new learning (*Rescorla and Wagner, 1972*). During fear extinction, the omission of the aversive US is an

unexpected outcome which initiates new learning about the CS, specifically that it no longer predicts danger. However, how this learning is initiated at the neuronal level has remained unknown. Here, by using bidirectional optogenetic manipulations, we demonstrated that the DA signal during the omission of the aversive US drives normal fear extinction learning: inhibiting this DA signal prevented, while enhancing it accelerated, normal extinction learning. Our results are consistent with previous findings establishing the causal role of DA neurons in reinforcement learning (*Chang et al., 2016*; *Steinberg et al., 2013*; *Tsai et al., 2009*) and further extend their role to safety learning. Consistent with our findings, a recent study inhibited VTA DA neurons during US omission in rats and also found reduced fear extinction learning (*Luo et al., 2018*). Our study replicates this finding in mice and further extends it by showing that enhancing DA neuron activity at the time of the US omission is also sufficient to accelerate normal extinction learning. Consistent with our results, it has been shown that enhancement of DA signaling by L-DOPA administration during extinction was sufficient to initiate fear extinction learning in a mouse model of impaired extinction learning (*Whittle et al., 2016*). Our results therefore suggest that enhancement of DA signaling during extinction could be a potential strategy for the treatment of anxiety disorders.

Fear extinction is mediated by a network of brain structures consisting mainly of the amygdala and the infra-limbic (IL) subregion of the medial prefrontal cortex (*Duvarci and Pare, 2014*; *Maren et al., 2013*; *Pape and Pare, 2010*; *Sotres-Bayon and Quirk, 2010*; *Tovote et al., 2015*). Although neuronal activation has been observed in these structures during different stages of fear extinction, they occur during the CS and later than the DA signal we observed in our study. In the basolateral amygdala, a subpopulation of neurons termed 'extinction neurons' increases their firing to the CS during extinction learning (*Amano et al., 2011*; *Herry et al., 2008*). However, these neurons become CS responsive late in the extinction session right before the animals show a decrease in fear responses (*Herry et al., 2008*) suggesting that these neurons likely mediate inhibition of fear responses. In the IL, a structure necessary for consolidation of extinction memories (*Sotres-Bayon and Quirk, 2010*), increased firing to the CS is observed during extinction recall (*Milad and Quirk, 2002*). In contrast, the DA signal that we demonstrated in our study occurs at the early trials of the extinction session, supporting our conclusion that this signal initiates extinction learning.

Plasticity in the amygdala and IL underlie acquisition and consolidation of fear extinction memories (*Duvarci and Pare, 2014*; *Maren et al., 2013*; *Pape and Pare, 2010*; *Sotres-Bayon and Quirk, 2010*; *Tovote et al., 2015*). Furthermore, DA signaling in the amygdala and IL has been shown to play an important role in fear extinction (*Abraham et al., 2014*; *Haaker et al., 2013*; *Hikind and Maroun, 2008*; *Mueller et al., 2010*; *Shi et al., 2017*; *Whittle et al., 2016*). Interestingly, optogenetic inhibition of DA neurons during US omission has been found to prevent extinction-related plasticity in the amygdala and IL (*Luo et al., 2018*). However, it is unclear how this DA signal induces plasticity in the amygdala and IL to underlie extinction memory. This can be mediated through direct DA projections to these structures or indirectly through a multi-synaptic circuit mechanism. The first step in addressing this issue will be to identify the projection target of these DA neurons that signal the omission of the US during fear extinction.

Recent studies have shown that midbrain DA neurons form functionally distinct and mostly non-overlapping subpopulations based on their projection targets (*Beier et al., 2015*; *Lammel et al., 2008*; *Lammel et al., 2011*; *Lerner et al., 2015*; *Lynd-Balta and Haber, 1994*; *Menegas et al., 2015*; *Menegas et al., 2017*; *Parker et al., 2016*; *Roeper, 2013*). Therefore, an important question is which subpopulation of VTA DA neurons generates the response to the US omission. One possible candidate is the subpopulation projecting to the NAc. Supporting this possibility, an increase in DA release around the time of the CS offset during fear extinction has been observed in the NAc (*Badrinarayan et al., 2012*). In particular, this was observed only during the early trials of extinction, consistent with our results. DA signaling in the NAc has also been shown to be important for relief learning (*Mayer et al., 2018*) and avoidance behavior (*Gentry et al., 2016*; *Oleson et al., 2012*). Whether fear extinction learning and relief and avoidance learning share related mechanisms and involve overlapping subpopulations of DA neurons projecting to NAc is not known and will be an important question for future research. Notably, pharmacological blockade of DA receptors in the NAc have been found to impair fear extinction learning (*Holtzman-Assif et al., 2010*). Furthermore, we have observed that fear extinction learning in humans is accompanied by a prediction error-like activation in the ventral striatum (*Raczka et al., 2011*). However, at odds with these findings, inhibition of DA terminals in NAc or DA neurons projecting to the medial shell of NAc during US omission

did not affect fear extinction learning, although it did impair consolidation of extinction memory, in a recent study (*Luo et al., 2018*). Our single unit results demonstrate that a small subpopulation of DA neurons mediate this DA signal to drive extinction learning. Therefore, it is possible that this sub-population of DA neurons projects to a specific subregion of NAc that was not targeted by *Luo et al. (2018)*. In addition to NAc, other possible candidates include the DA neurons that project to the amygdala and/or IL. Identifying which projection-defined subpopulation of DA neurons signals the omission of the US to initiate fear extinction learning will be an important question for future research.

In conclusion, our study identifies a prediction error-like signal encoded by DA neurons that is necessary to initiate fear extinction learning. Furthermore, we found that enhancing this DA signal is sufficient to accelerate extinction learning and strengthen extinction memory consolidation. Deficits in fear extinction learning are thought to underlie anxiety disorders (*Craske et al., 2017*; *Graham and Milad, 2011*; *Mahan and Ressler, 2012*; *Milad and Quirk, 2012*; *Pitman et al., 2012*). Our study therefore has therapeutic implications for anxiety disorders by identifying DA neuron activity as a potential target for novel treatments.

## Materials and methods

### Subjects

All procedures were conducted in accordance with the guidelines of the German Animal Protection Act and were approved by the local authorities (Regierungsprasidium Darmstadt; protocol number 1038). Male C57BL/6N mice (Charles River), aged 3 months at the start of experiments, were used in the in vivo electrophysiology experiment. Male heterozygous DAT-Cre mice (*Zhuang et al., 2005*; backcrossed with C57BL/6N) aged 3–6 months at the start of experiments were used in the photom-etry and optogenetics experiments. All experimental groups were matched for age. All mice were individually housed on a 12 hr light/dark cycle. All experiments were performed during the light cycle.

### Viruses

AAV5-EF1a-DIO-hChR2(H134R)-eYFP, AAV5-EF1a-DIO-eNpHR3.0-eYFP, AAV5-EF1a-DIO- eYFP and AAV5-CAG-Flex-GFP were produced and packaged by the University of North Carolina Vector Core. AAV5-CAG-Flex-GCaMP6f-WPRE-SV40 and AAV5-CAG-Flex- GCaMP6s-WPRE-SV40 were produced and packaged by the University of Pennsylvania Vector Core.

### Surgical procedures

Animals were anesthetized using isoflurane (1–2%) and placed in a stereotaxic frame. At the onset of anesthesia, all animals received subcutaneous injections of carprofen (4 mg/kg) and dexamethasone (2 mg/kg). The animal's temperature was maintained for the duration of the surgical procedure using a heating blanket. Anesthesia levels were monitored throughout the surgery and the concentration of isoflurane adjusted so that the breathing rate never fell below 1 Hz. After exposing the skull sur-face, craniotomies were made overlying the VTA (3.2 mm posterior to bregma and 0.5 mm lateral to the midline).

For in vivo single-unit recordings, we used a moveable bundle of 5–8 stereotrodes made by twist-ing together two 0.0005 inch tungsten wires (M219350, California Fine Wire). The stereotrode bun-dle was attached to a custom-made microdrive that made it possible to advance the electrodes along the dorsoventral axis. On the day of implantation, the stereotrodes were gold-plated to reduce the impedance to 0.2–0.3 MΩ at 1 kHz. The stereotrode bundle was inserted through the craniotomy above the VTA to a depth of 3.9–4.0 mm below bregma. All electrode wires were con-nected to an electrode interface board (EIB-16; Neuralynx) for relaying electrophysiological signals to the data acquisition system. The microdrive was anchored to the skull using skull screws and den-tal cement (Paladur).

For the photometry experiments, DAT-cre mice were injected unilaterally with 1 µl of AAV5-CAG-Flex-GCaMP6f-WPRE-SV40 (final titer $2.7 \times 10^{12}$ pp per ml) or AAV5-CAG-Flex- GCaMP6s-WPRE-SV40 (final titer $6.4 \times 10^{12}$ pp per ml) or AAV5-CAG-Flex-GFP (final titer $4.5 \times 10^{12}$ pp per ml). Viruses were injected in the VTA (3.2 mm posterior to bregma, 0.5 mm lateral to the midline and 4.5

mm ventral to bregma) at 50 nl/min using a 10 µl syringe with a 33-gauge needle controlled by an injection pump. The needle was left in place for an additional 10–15 min before slowly being withdrawn. Following infusion of the virus, an optical fiber (400 µm core diameter, 0.48 NA, Doric Lenses) was slowly inserted through the same craniotomy into the VTA at a depth of 4.1–4.3 mm below bregma. The optical fiber was then anchored to the skull using skull screws and dental cement (Paladur).

For optogenetic experiments, DAT-cre mice were injected bilaterally in the VTA with 1 µl of AAV5-EF1a-DIO-hChR2(H134R)-eYFP (final titer $4.3 \times 10^{12}$ pp per ml), AAV5-EF1a-DIO- eNpHR3.0-eYFP (final titer $4 \times 10^{12}$ pp per ml) or AAV5-EF1a-DIO-eYFP (final titer $4.4 \times 10^{12}$ pp per ml) per hemisphere using the coordinates described above. Optical fibers (200 µm core diameter, 0.22 NA, Thorlabs) were implanted bilaterally above the VTA to a depth of 3.9–4.0 mm below bregma as described above.

## Behavior

Fear conditioning and extinction took place in two different contexts (A and B). Context A consisted of a square chamber with an electrical grid floor (Med Associates) used to deliver footshock. Context B consisted of a white teflon cylindrical chamber with bedding material on the floor. The chambers were located inside a sound attenuating box and were cleaned with 50% ethanol and 1% acetic acid before and after each session. All mice were habituated to contexts A and B for 10–15 min each in a counterbalanced fashion. For electrophysiology and photometry experiments, mice received a tone habituation session followed by fear conditioning on day 1. Tone habituation started following a 2 min baseline period in context A and consisted of 10 presentations of the CS (4 kHz tone, 75 dB, 10 s) with a random intertrial interval (ITI) of 40–120 s. Fear conditioning consisted of five pairings of the CS with a US (1 s footshock, 0.55 mA, ITI: 40–120 s). The onset of the US coincided with the offset of the CS. On day 2, mice received an extinction session consisting of 25–30 and 30 presentations of the CS alone in context B in the electrophysiology and photometry experiments, respectively. For optogenetic experiments, on day 1 mice underwent fear conditioning consisting of four pairings of a CS (4 kHz tone, 75 dB, 30 s) with a US (1 s footshock) with a random ITI of 40–120 s in context A. The US intensity was 0.45 mA and 0.5 mA in the optogenetic inhibition and excitation experiments, respectively. On day 2, mice received an extinction session consisting of 25 presentations of the CS alone in context B. Twenty-four hours later, mice received an extinction recall test in context B. The extinction recall test consisted of 25 and 10 presentations of the CS alone in the optogenetic inhibition and excitation experiments, respectively.

The behavior of mice was recorded to video and scored by an experienced observer blind to the experimental condition. Behavioral freezing, defined as the absence of all bodily movements except breathing-related movement (*Blanchard and Blanchard, 1972*), was used as the measure of fear. Animals that showed low conditioned fear (<50% freezing) at the start of the extinction session (First CS of extinction) were excluded from the study. This criterion led to the exclusion of 1 mouse from photometry and one mouse from optogenetics experiments.

## In vivo single-unit electrophysiology

Following one week of recovery from surgery, animals were habituated to handling and moving with the wire tether connecting the microdrive to the recording system. Before fear conditioning began, stereotrodes were slowly advanced until single-units with low baseline firing rate (<10 Hz) were observed, to increase the probability of recording putative DA neurons during the task. The spike waveforms tended to change from day 1 (tone habituation and conditioning) to day 2 (extinction) even when electrodes were not moved or in case cells were lost overnight, we advanced (40–80 µm) the microdrive to find new cells on day 2. We therefore treated the cell populations recorded on these two days as independent. On each day of the fear conditioning task, single-units were first recorded for 5–10 min while animals were in their homecage to assess their baseline firing rates.

Neural data were acquired using a 16-channel headstage (HS-18, Neuralynx) that was connected to the electrode interface board and relayed the signals to a Digital Lynx SX (Neuralynx) data acquisition system. To extract putative spikes, neural signals were bandpass filtered between 600 and 6000 Hz, and waveforms that passed the threshold (50–60 µV) were digitized at 30 kHz. One stereotrode channel that did not have any apparent units was used as reference. We also recorded the

neural signals bandpass filtered between 1 and 6000 Hz to analyze the spike waveforms. In order to verify recording locations, current (50 mA, 10 s) was passed through one of the stereotrode channels to produce a lesion in the recording site at the end of the experiment.

## Analysis of single-unit data

Spike waveforms were sorted offline into single-unit clusters using SpikeSort3D (Neuralynx). Only well-isolated single-units that displayed a clear refractory period (>1 ms) were included in subsequent analysis, performed using scripts custom-written in MATLAB (MathWorks). The VTA contains different cell types, including DA neurons and gamma-aminobutyric acid (GABA) neurons (*Morales and Margolis, 2017*). Midbrain DA neurons typically exhibit baseline firing rates below 10 Hz in awake animals (*Jin and Costa, 2010*; *Li et al., 2012*) and long duration action potentials (*Grace and Bunney, 1980*; *Jin and Costa, 2010*; *Li et al., 2012*). We therefore classified neurons whose baseline firing rate (measured in the home cage) were below 10 Hz and exhibited long duration action potentials (peak-to-peak duration >450 μs) as putative DA neurons (*Figure 1—figure supplement 3*). Spike waveforms were analyzed using the neural signals bandpass filtered between 1 and 6000 Hz. To analyze firing rates during the task, we constructed peri-stimulus time histograms (PSTHs) aligned to the onset of the CS (−5 s to +15 s) using 1 s bins. PSTHs were calculated separately for early extinction trials (E-Ext: average of first 10 CSs), late extinction trials (L-Ext: average of last 10 CSs) and tone habituation (Hab: average of 10 CSs). These PSTHs were then normalized with a z-score transformation by subtracting the baseline firing rate (5 s pre-CS period) from each individual 1 s bin and then dividing this difference by the standard deviation of the baseline. A neuron was classified as US omission excited if it met two criteria: 1) z-score was greater than 2 at the time of the US omission (during the 1 s bin following the offset of the CS); 2) the z-score had to increase by at least two from the last bin of the CS to the US omission bin to ensure that the increase in firing was specific to the omission of the US and not a result of sustained increase in firing to the CS. Conversely, a neuron was classified as US omission inhibited if (1) z-score was smaller than −2 at the time of the US omission (during the 1 s bin following the offset of the CS); (2) the z-score had to decrease by at least two from the last bin of the CS to the US omission bin to ensure that the decrease in firing was specific to the omission of the US and not a result of sustained decrease in firing to the CS. To quantify responses to the CS, we averaged the z-scores during the entire CS period (10 s) for Hab, E-Ext and L-Ext trials and obtained an average z-score for each neuron. The neurons were classified as excited or inhibited by the CS if they showed average z-scores greater than two or smaller than −2, respectively.

## Calcium recordings using fiber photometry

Animals were injected with viral vectors and implanted with an optical fiber in the VTA, as described above. After a waiting period of 3–4 weeks to allow for surgical recovery and virus expression, animals were connected to a 400 μm patch cord (Doric Lenses). Fluorescence was measured by delivering 465 nm excitation light through the patch cord and separating the emission light at 525 nm with a beamsplitter (Fluorescence MiniCube FMC5, Doric Lenses). The emission light was collected using a Femtowatt Silicon Photoreceiver (Model # 2151, Newport). The voltage output of the photoreceiver was then digitized at 2 kHz (Digital Lynx SX, Neuralynx). Using the same approach we also measured the isosbestic (activity-independent) fluorescence of gCaMP6 at 430 nm using 405 nm excitation light. After animals were habituated to handling and being connected to the patch cord, they underwent the fear conditioning protocol. At the end of fear conditioning experiments, animals were tested on an operant conditioning task (*Figure 3—figure supplement 2*). To this end, animals were placed in an operant chamber containing a water delivery port. Each nosepoke into the delivery port that followed the previous nosepoke by a variable inter-trial interval (3–5 s) triggered the delivery of liquid reward (10% sucrose solution) with a 50% probability. This task was used to verify that reward delivery caused large increases in fluorescence in gCaMP6-expressing mice, as expected based on previous electrophysiology (*Bayer and Glimcher, 2005*; *Eshel et al., 2015*; *Eshel et al., 2016*; *Roesch et al., 2007*; *Schultz et al., 1997*) and fiber photometry (*Menegas et al., 2017*; *Parker et al., 2016*; *Soares et al., 2016*) studies, suggesting that this $Ca^{+2}$ signal is indeed generated by DA neuron activity.

## Analysis of fiber photometry data

The voltage output of the photoreceiver, representing fluctuations in fluorescence, was first low-pass filtered at 4 Hz and then downsampled to 10 Hz. The change in fluorescence evoked by the CS (dF/F) was then calculated by subtracting from each trace the baseline fluorescence (average during the 5 s before CS onset) and dividing it by the baseline fluorescence. dF/F traces were then averaged separately for each animal for early extinction trials (E-Ext: average of first 10 CSs), late extinction trials (L-Ext: average of last 10 CSs) and tone habituation (Hab: average of 10 CSs). To examine responses to US omission we further averaged dF/F values in the 5 s following CS offset. Similar results were obtained from animals expressing gCaMP6f (n = 5) and gCaMP6s (n = 5), therefore the data was pooled. To quantify responses to the CS, we averaged the dF/F values during the CS for each animal for Hab, E-Ext and L-Ext. To quantify responses to reward (*Figure 3—figure supplement 2*), average dF/F was calculated separately for rewarded and unrewarded nosepokes using the baseline fluorescence 3 s before noseport entry.

## Optogenetic manipulations

For bilateral optogenetic manipulations during behavior, the implanted optical fibers (200 µm core diameter, 0.22 NA, Thorlabs) were connected to 200 µm patch cords (Thorlabs) with zirconia sleeves and the patch cords were connected to a light splitting rotary joint (FRJ 1 × 2 i, Doric Lenses) that was connected to a laser with a 200 µm patch cord (Thorlabs). For mice expressing the light-activated inhibitory opsin halorhodopsin (NpHR-eYFP) and their eYFP controls, yellow light was delivered from a DPSS 594 nm laser (Omicron). Laser power at the tip of the optic fiber was 10–15 mW. To inhibit DA neurons around the time of the US omission during the extinction session on day 2, the laser was turned on from 1 s before to 2 s after the CS offset. The laser was then turned off gradually using a 1 s ramp to avoid rebound excitation (*Mahn et al., 2016*; *Figure 5—figure supplement 2*). For mice in the Unpaired-NpHR control group, the light was delivered the same way during the ITIs. For mice expressing the light activated excitatory opsin channelrhodopsin-2 (ChR2) and their eYFP controls, blue light pulses were delivered from a 473 nm laser (LuXx473, Omicron). The laser power at the tip of the optic fiber was 5–10 mW. To phasically activate DA neurons at the time of the US omission, 5 ms light pulses were delivered at 20 Hz for 1 s beginning at the offset of the CS. The mice in the Unpaired-ChR2 group received delivery of the same light pulses during the ITIs.

## In vivo single unit recordings during optogenetic excitation and inhibition

Using the surgical procedures described above DAT-cre mice were injected in the VTA with 1 µl of AAV5-EF1a-DIO-hChR2(H134R)-eYFP (final titer 4.3 × $10^{12}$ pp per ml) or AAV5-EF1a-DIO-eNpHR3.0-eYFP (final titer 4 × $10^{12}$ pp per ml) per hemisphere using the coordinates described above. Two to three weeks after virus injection, animals were anesthetized and placed in a stereotaxic frame with the skull exposed. A stainless-steel head post (Luigs and Neumann) was then cemented to the exposed skull. The area of the skull overlying VTA was left free of cement but covered with a silicon elastomer (Kwik-Sil, World Precision Instruments). Skull screws were inserted over the frontal cortex and cerebellum to serve as reference and ground, respectively, and to provide anchoring support for the cement. Following recovery from surgery, animals were handled and habituated to being head-fixed, by inserting the head post into a matching head post holder (Luigs and Neumann). Following 2–3 days of habituation to being head-fixed, animals underwent another surgery to prepare a craniotomy over the VTA. Animals were anesthetized with isoflurane and placed in a stereotaxic frame. The Kwik-Sil was removed from the skull and a small craniotomy was then made in the skull over the VTA and then sealed with Kwik-Sil. The following day, the animals were head-fixed, the Kwik-Sil removed, and a 32-channel optrode (silicon probe with an optical fiber; A1 × 32-Edge-5mm-20–177-OA32, NeuroNexus) was lowered into the VTA using a micromanipulator (SM-8, Luigs and Neumann). The optrode was then advanced to a depth of ~4700 µm below the brain surface to span the dorsal-ventral extend of the VTA. Following final placement of the optrode and a brief waiting period (~15 min), neural activity was recorded while laser pulses (yellow light: 10–15 mW, blue light: 5–10 mW) were delivered through a patch cord (Neuronexus) connected to the optic fiber. Electrophysiological signals were filtered between 1 and 6000 Hz, digitized at 30 kHz using a digitizing headstage (RHD2132 Amplifier Board, Intan Technologies), and acquired using a

USB interface board (RHD2000, Intan Technologies). Once the recording was over, optrode was removed from the brain and then was penetrated to another location in the VTA. Multiple penetrations and recordings were performed during a session. Before the final penetration, the silicone probe was coated with a fluorescent dye (DiI, Life Technologies) to assist with the identification of the recording location.

## Histology

At the end of the experiments, mice were deeply anesthetized with sodium pentobarbital and were transcardially perfused with 4% paraformaldehyde and 15% picric acid in phosphate-buffered saline (PBS). Brains were removed, post-fixed overnight and coronal brain slices (60 μm) were sectioned using a vibratome (VT1000S, Leica). Standard immunohistochemical procedures were performed on free-floating brain slices. Briefly, sections were rinsed with PBS and then incubated in a blocking solution (10% horse serum, 0.5% Triton X-100% and 0.2% BSA in PBS) for 1 hr at room temperature. Slices were then incubated in a carrier solution (1% horse serum, 0.5% Triton X-100% and 0.2% BSA in PBS) containing the primary antibody overnight at room temperature. The next day, the sections were washed in PBS and then incubated in the same carrier solution containing the secondary antibody overnight at room temperature. The following primary antibodies were used: polyclonal rabbit anti-tyrosine hydroxylase (TH, catalog # 657012, 1:1000, Calbiochem), monoclonal mouse anti-TH (catalog # MAB318, 1:1000, Millipore), polyclonal rabbit anti-GFP (catalog # A11122, 1:1000, Life Technology). The following secondary antibodies were used: Alexa Fluor 568 goat anti-rabbit (catalog # A11011, 1:1000, Thermo Fisher Scientific, Invitrogen), Alexa Fluor 568 goat anti-mouse (catalog # A11004, Thermo Fisher Scientific, Invitrogen), and Alexa Fluor 488 goat anti-rabbit (catalog # A11008, 1:1000, Thermo Fisher Scientific, Invitrogen). Finally, sections were washed with PBS, mounted on slides and coverslipped with a 4',6-diamidin-2-phenylindol (DAPI) containing medium (VECTASHIELD, Vector Laboratories) or incubated for 10 min in 0.1 M PBS containing 0.02% DAPI (catalog # D1306, Molecular Probes, Invitrogen), washed for 10 min in PBS, mounted on slides and coverslipped. Animals with incorrect electrode/optical fiber placements were excluded from data analysis. A total of four mice were excluded from the study due to incorrect placement of electrodes/optical fibers. We excluded one animal from the electrophysiology experiment because the electrodes were placed outside (posterior to) the VTA. Three animals were excluded from the fiber photometry experiment. In one animal the optical fiber was placed too dorsal and in the other two animals too ventral in the VTA. In these animals, the neuronal activity dependent fluorescence signal was undetectable or too low. No animals were excluded from optogenetic experiments.

## Statistics

Data were statistically analyzed using GraphPad Prism (GraphPad Software) and MATLAB (Mathworks). All statistical tests were two-tailed and had an α level of 0.05. All error bars show s.e.m. All ANOVAs were followed by Bonferroni *post hoc* tests if significant main or interaction effects were detected. No statistical methods were used to predetermine sample size, but our sample sizes were similar to those generally used in the fear conditioning field. Animals were randomly assigned to experimental groups before the start of each experiment after ensuring that all experimental groups were matched for age. All results were obtained using groups of mice that were run in several cohorts.

## Acknowledgements

We would like to thank Jochen Roeper for helpful discussions; and Beatrice Fischer, Jasmine Sonntag and Thomas Wulf for technical assistance. We thank René Hen for generating, and Jochen Roeper and Institute of Neurophysiology for supporting the breeding of and providing DAT-cre mice. This work was supported by the Deutsche Forschungsgemeinschaft (DFG SFB1193 Grant INST247/852-1 to SD and INST247/854-1 to TS).

## Additional information

### Funding

| Funder | Grant reference number | Author |
| --- | --- | --- |
| Deutsche Forschungsgemeinschaft | SFB1193 Grant INST247/852-1 | Sevil Duvarci |
| Deutsche Forschungsgemeinschaft | SFB1193 Grant INST247/854-1 | Torfi Sigurdsson |

The funders had no role in study design, data collection and interpretation, or the decision to submit the work for publication.

### Author contributions

Ximena I Salinas-Hernández, Data curation, Formal analysis, Investigation, Methodology, Writing—review and editing; Pascal Vogel, Data curation, Software, Formal analysis, Methodology; Sebastian Betz, Software, Methodology; Raffael Kalisch, Conceptualization, Writing—review and editing; Torfi Sigurdsson, Resources, Software, Formal analysis, Funding acquisition, Methodology, Writing—original draft, Writing—review and editing; Sevil Duvarci, Conceptualization, Resources, Data curation, Software, Formal analysis, Supervision, Funding acquisition, Investigation, Methodology, Writing—original draft, Project administration, Writing—review and editing

### Author ORCIDs

Sevil Duvarci http://orcid.org/0000-0003-4494-7741

### Ethics

Animal experimentation: All procedures were conducted in accordance with the guidelines of the German Animal Protection Act and were approved by the local authorities (Regierungsprasidium Darmstadt; protocol number 1038). Every effort was made to minimize suffering.

### Decision letter and Author response

Decision letter https://doi.org/10.7554/eLife.38818.023
Author response https://doi.org/10.7554/eLife.38818.024

## Additional files

### Supplementary files

• Transparent reporting form
DOI: https://doi.org/10.7554/eLife.38818.021

### Data availability

All data generated or analysed during this study are included in the manuscript and supporting files.

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
