## [Decision Letter]

Thank you for submitting your article "Dopamine Neurons Drive Fear Extinction Learning by Signaling the Omission of Expected Aversive Outcomes" for consideration by *eLife*. Your article has been reviewed by three peer reviewers, and the evaluation has been overseen by a Reviewing Editor and a Senior Editor. The following individuals involved in review of your submission have agreed to reveal their identity: Stephan Lammel (Reviewer #1); Naoshige Uchida (Reviewer #2); Nicolas Singewald (Reviewer #3).

The reviewers have discussed the reviews with one another and the Reviewing Editor has drafted this decision to help you prepare a revised submission.

The reviewers see value in your work, and think it is interesting, but also think that more attention needs to be placed in describing what is novel in this study versus other recent studies.

In addition, the reviewers request more details and potential experiments, especially regarding the electrophysiological recordings. They suggest further characterization of the electrophysiological recordings (with potentially an increase of N of neurons, which is quite low), more details about the classification of the neurons (and photoId if available), characterization of the responses of putative non-DA neurons, and if possible the characterization of the effects of the optogenetic manipulations.

Below you find the point-by-point comments that can help you prepare your revision.

Reviewer #1:

In this study, Salinas-Hernandez et al. investigate DA neurons during fear extinction learning and describe a role of these cells for "safety learning". Major findings of this study include: i) Midbrain DA neurons are activated by the omission of an aversive stimuli during fear extinction and ii) optogenetic inhibition or stimulation of DA neurons at the time when an expected aversive stimulus is omitted impairs or accelerates fear extinction learning, respectively.

The results in this manuscript are convincing, the experiments are well done and I do not have any major technical concerns. However, a major concern is that, at least in its current form, the study provides relatively little novel insights and no substantial conceptual advance. The concept that omission of an expected aversive outcome activates DA neurons and increases DA release in the NAc during extinction is well established. This has been shown using various methodological approaches including in vivo electrophysiology and fast scan voltammetry (Badrinarayan et al., 2012; Matsumoto and Hikosaka, 2009; Oleson et al., 2012). This concept is consistent with previous studies reporting that the increase in DA firing or DA release caused by an aversive stimulus may reflect the rewarding effects after its termination or the relief caused when it is omitted – a process which is probably very similar to extinction (Brischoux et al., 2009; Budygin et al., 2012; Mirenowicz and Schultz, 1996). The finding that DA activity during the omission of an expected aversive outcome is necessary for fear extinction learning seemed to be the most exciting finding of this paper. However, I noticed that the same finding has been published just recently by another group (Luo et al., 2018). In addition, the Johansen group presented evidence that this effect is specifically mediated by NAc-projecting DA neurons – a finding that I felt was missing in the present work. So, it's nice to see replication of this previous work, but the reviewing editors will need to decide how to weigh reproducibility vs. novelty in this case.

What was surprising to me is that the authors only found a very small number of putative DA neurons (8/30) that exhibit an increase in firing rate to the omission of the aversive US during the early extinction trials (subsection “Dopamine Neurons Signal the Unexpected Omission of the Aversive US during Fear Extinction Learning”, first paragraph and Figure 1). Let's assume the recorded cells are indeed all dopaminergic, then it means that only a very small subpopulation of DA neuron mediates this effect. A limitation of the fiber photometry approach is that there is little information about how uniform the activity is across neurons and changes in net activity may be the dominant pattern of activity of a small subset of neurons. Additionally, the data in Figure 2F suggest that at least some animals also showed an increase in calcium activity during the CS presentation.

It is also possible that the non-responsive cells in Figure 1 are non-dopaminergic (i.e., putative GABAergic or glutamatergic neurons). Thus, another limitation of the present work is that the authors did not unambiguously identify DA neurons. Their conclusions rely upon indirect methods such as spike waveforms and response to rewarding stimuli. These identification methods have been called into question and may not be appropriate for the identification of DA neurons (Margolis et al., 2006, 2010; Ungless and Grace, 2012). Thus, the authors should consider alternative approaches such as an optogenetic tagging method to unambiguously identify DA neurons (and other VTA cell populations) in highly heterogeneous brain regions such as the VTA – they clearly have the technical expertise to do this.

References:

Brischoux, F., Chakraborty, S., Brierley, D.I., and Ungless, M.A. (2009). Phasic excitation of dopamine neurons in ventral VTA by noxious stimuli. Proc. Natl. Acad. Sci. 106, 4894-4899.

Budygin, E.A., Park, J., Bass, C.E., Grinevich, V.P., Bonin, K.D., and Wightman, R.M. (2012). Aversive stimulus differentially triggers subsecond dopamine release in reward regions. Neuroscience 201, 331-337.

Margolis, E.B., Lock, H., Hjelmstad, G.O., and Fields, H.L. (2006). The ventral tegmental area revisited: is there an electrophysiological marker for dopaminergic neurons? J Physiol 577, 907-924.

Margolis, E.B., Coker, A.R., Driscoll, J.R., Lemaître, A.-I., and Fields, H.L. (2010). Reliability in the identification of midbrain dopamine neurons. PloS One 5, e15222.

Mirenowicz, J., and Schultz, W. (1996). Preferential activation of midbrain dopamine neurons by appetitive rather than aversive stimuli. Nature 379, 449-451.

Ungless, M.A., and Grace, A.A. (2012). Are you or aren't you? Challenges associated with physiologically identifying dopamine neurons. Trends Neurosci 35, 422-430.

Reviewer #2:

It has been proposed that fear extinction induces new learning, not merely the erasure of the previous fear memory. The mechanism underlying new learning during fear extinction remains elusive. In this study, Salinas-Hernandez and colleagues examined the role of VTA dopamine neurons in this process. The authors first characterized the activity of VTA dopamine neurons using electrophysiological recording and fiber photometry during fear extinction. The authors show that 26.6% of putative dopamine neurons were excited by the omission of electrical shock during the early extinction period, while a smaller fraction of neurons (10%) were excited during the late extinction period. Furthermore, the authors show that optogenetic inhibition of VTA dopamine neurons at the time of omission of electric shock impaired fear extinction as measured by the decrease of freezing. The authors also show that optogenetic activation of VTA dopamine neurons at the time of omission of electric shock facilitated the decrease of freezing.

Previous studies, using cyclic voltammetry and pharmacology/lesions, have indicated the role of dopamine in fear extinction. The present study extends these studies significantly by reporting single dopamine neuron activities, and by examining the effect of manipulating the activity of VTA dopamine neurons precisely at the time of shock omission using optogenetics. The results of these experiments are convincing and somewhat surprising. This study provides important insights into the mechanism of fear extinction learning.

There are some recent publications that addressed similar questions independently (Luo et al., 2018; Mayer et al., 2018), which reached overlapping conclusions. The present study complements these studies by providing additional information which were not reported in other studies (e.g. single neuron recording and the effect of optogenetic stimulation). Overall, these studies together make important contributions to the fear extinction literature.

1) One of the important and novel aspects of this study is single neuron recording during fear extinction. Although the authors report the minimum results to support their conclusions, this study can be further strengthened if the authors report a little more detail of the recorded neuron population. The authors first select those that were activated during shock omission and report the average firing pattern of these neurons. This represents only a fraction of all putative dopamine neurons (8 out of 30 putative dopamine neurons). It is important to know how other putative dopamine neurons responded (no response or were some of them inhibited?). Please show the distribution of the responses of all putative dopamine neurons during shock omission. Is the average response of all putative dopamine neurons significantly positive?

Furthermore, the authors have recorded the activity of unidentified dopamine neurons. Although this is not essential for the authors' conclusion, it would be useful to report how these neurons responded. Are there narrow-spiking neurons that were activated during shock omission? This may have already been done in Figure 1—figure supplement 2: does the red circle represent a putative non-dopaminergic neuron that was activated during shock omission? Do the black circles actually represent putative non-dopaminergic neurons that were not activated during shock omission? Please clarify. Also, it would be useful to describe these results in the main text.

2) Please report the result of quantification for Figure 2B, 3B, C (TH staining compared with the expression of NpHR or ChR2).

3) Figure 2G (the correlation between 'Percent freezing' and 'Average dF/F') is very interesting. However, additional information is needed to interpret this result. First, it is better to use the reduction in freezing rather than the absolute value of freezing, in order to represent learning. Does the correlation hold even if the authors use the difference of freezing between before and during/after extinction? Second, it is a little unclear what is causing the variation in the photometry signal. One possibility, which the authors would like to exclude, is the variation in recording location. To address this, can the authors examine the correlation between freezing (ideally the change in freezing) and reward response? Not obtaining significant correlation in this analysis would strengthen the authors' conclusion.

4) Please cite and discuss Luo et al., 2018 and Mayer et al., 2018.

Reviewer #3:

Combining sophisticated electrophysiological, photometric and optogenetic techniques the authors studied the question of how dopaminergic neurons are involved in extinction learning. In their experiments they were able to demonstrate the importance of activity of the midbrain ventral tegmental area (VTA) dopamine system in the initiation and facilitation of cued fear extinction. Activation of dopaminergic neurons is observed when an expected aversive event does not occur, which is a crucial phase in the onset of extinction learning. The study is exciting and in large parts novel. It also provides further information on the neuronal basis of findings of a very recently published study by Luo et al. reporting a role for the VTA-nucleus accumbens dopamine system in extinction learning. Methods are state-of-the art, experiments were carried out with care and the manuscript is straightforward to read. However, there are points that need to be addressed.

1) As mentioned above, in a paper published a few weeks ago Luo and co-workers demonstrate that the optogenetic inactivation of VTA-dopamine cells during the US omission period reduced extinction learning in rats. Here, part of this finding is replicated in another species and extended as, for example: a) the optogenetic stimulation of VTA dopaminergic activity facilitated fear extinction in mice (Figure 4); b) (Re)extinction is possible in the paired-NpHR mice during extinction recall when the laser is switched off (Figure 3). This finding may also suggests that silencing of the VTA-dopamine activity did not reconsolidate the fear memory. It is necessary to cite the Luo paper and emphasize the overlap and differences (some of which mentioned above) with the present manuscript.

2) Please provide evidence that optical silencing and stimulation using NpHR and ChR2, respectively, is a valid method to modulate dopaminergic activity in the VTA of mice.

3) The dopamine system has been suggested to be involved in both the learning and consolidation of fear extinction. The authors nicely demonstrate its role in the initiation of fear extinction learning, but unfortunately do not further discuss this finding in relation to published work. For example, it has been shown that administration of the dopaminergic drug L-DOPA before extinction training was able to initiate fear extinction learning in a mouse model of impaired fear extinction. This discussion needs to be added. Also the consolidation aspect is not addressed sufficiently (one mentioning that "extinction memory was strengthened"). Furthermore, the authors focus on the better-than-expected outcome reward aspect of dopamine in fear extinction in the Discussion. There is increasing evidence that dopamine activity/release may also reflect a prediction error signal that guides extinction learning. Whether and how the present data support this latter idea, is missing and should be added.

---

## [Author Response]

Reviewer #1:In this study, Salinas-Hernandez et al. investigate DA neurons during fear extinction learning and describe a role of these cells for "safety learning". Major findings of this study include: i) Midbrain DA neurons are activated by the omission of an aversive stimuli during fear extinction and ii) optogenetic inhibition or stimulation of DA neurons at the time when an expected aversive stimulus is omitted impairs or accelerates fear extinction learning, respectively.The results in this manuscript are convincing, the experiments are well done and I do not have any major technical concerns. However, a major concern is that, at least in its current form, the study provides relatively little novel insights and no substantial conceptual advance.

We thank the reviewer for a detailed reading of our manuscript. Although the reviewer finds our results convincing, his major concern is that “the study provides relatively little novel insights and no substantial conceptual advance”. We respectfully disagree with this assessment. To begin with we would like to emphasize that the key goal of our study was to investigate the neural mechanisms underlying the initiation of fear extinction learning. As we explained in the Introduction of the manuscript, this is a key outstanding question in the fear extinction literature. Because extinction learning happens when an expected aversive US does not occur, we hypothesized that dopamine (DA) neurons would be involved since they are known to respond to better-than-expected outcomes. Supporting this hypothesis, we found that DA neurons increased their activity at the time of the US omission during fear extinction and that this response correlated with extinction learning. Furthermore, using bidirectional optogenetic manipulations, we showed that this DA signal is not only necessary for, but also sufficient to accelerate, extinction learning. Our study therefore suggests that DA neurons initiate fear extinction by signaling US omission and makes an important contribution to our understanding of this form of safety learning.

As we acknowledged in the original manuscript, previous studies have demonstrated responses of DA neurons to the omission of an aversive stimulus. It appears that the reviewer’s assessment of the novelty of our results is to a large extent based on these studies (discussed below). However, we wish to emphasize that these previous studies do not clearly demonstrate that US omission responses are involved in fear extinction learning. In fact, omission responses have mainly been observed (although not always consistently) in studies using partial reinforcement paradigms (where the US is omitted on a subset of trials), which is not the same as extinction learning. Furthermore, two studies (Badrinarayan et al., 2012; Oleson et al., 2012) that examined DA release during fear extinction do not specifically examine or make any claims about the role of US omission responses in fear extinction learning (and in fact the second of these studies does not observe US omission responses at all). Finally, we acknowledge the study of Luo et al., 2018, that was published after our original submission and which found, as we did, that inhibiting DA neuron activity during US omission impairs fear extinction. However, our study also has numerous additional and novel findings that complement those of Luo et al., 2018. Taken together, we therefore believe that our study makes a novel and important contribution to the fear extinction literature. Because the reviewer brings up these studies in his following comments, we will discuss them and their relationship to our study in more detail below.

The concept that omission of an expected aversive outcome activates DA neurons and increases DA release in the NAc during extinction is well established. This has been shown using various methodological approaches including in vivo electrophysiology and fast scan voltammetry (Badrinarayan et al., 2012; Matsumoto and Hikosaka, 2009; Oleson et al., 2012).

The reviewer states that “The concept that omission of an expected aversive outcome activates DA neurons and increases DA release in the NAc during extinction is well established”. We respectfully disagree with the reviewer. First, the relatively few studies that have investigated the firing of DA neurons during omission of an expected aversive outcome have not always yielded consistent results. For instance, whereas Matsumoto and Hikosaka, 2009, found a small increase in firing to the omission of an aversive air-puff, Tian and Uchida, 2015, did not find any change in firing to air-puff omission. A more recent study, on the other hand, found a significant yet small increase in DA firing to air-puff omission when animals were tested in a low, but not in a high, reward context (Matsumoto et al., 2016) suggesting that the valence of the testing context influences omission responses. Finally, Menegas et al., 2017, did not observe any response to air-puff omission at the DA neuron terminals in the ventral striatum. These studies and their methodological differences to ours were discussed in the original manuscript. Perhaps the most important difference is that the behavioral paradigm used in these studies was not fear extinction but rather partial reinforcement, in which the expected aversive air-puff was omitted on a subset of trials (10-75%, depending on the study). Whether there was any extinction of behavioral responses to the stimulus predicting aversive airpuff in these studies is not clear. In fact, partial reinforcement has been shown to cause learning that is resistant to extinction, a phenomenon termed as “the partial reinforcement extinction effect” (Skinner, 1938; Bouton et al., 2014; Grady et al., 2016). Thus, although the results from partial reinforcement studies are interesting and relevant, they do not address the key question of our study, namely whether US omission responses of DA neurons are involved in fear extinction learning.

More directly relevant to our study are two previous studies that examined DA release in the nucleus accumbens (NAc) during conditioned fear expression and extinction using cyclic voltammetry. Badrinarayan et al., 2012, did indeed find that DA release in the NAc was significantly elevated compared to pre-CS levels in the time period after CS offset during extinction. However, it is unclear whether this was specific to the US omission period since DA release appears to start increasing during the CS (see their Figures 3B and 3H). Indeed, the authors themselves refer to this response “as a diffuse increase because it was not nearly as robustly or reliably associated with any temporal attributes of cue presentation” (p. 15783). Although they consider the possibility that the DA signal at CS offset could be a ‘relief signal’ they also acknowledge that “there is a gradual but not significant increase to that point throughout tone presentation that would not be explained by a relief signal (Figure 3B, H)” (p. 15788). Notably, Badrinarayan et al. (2012) do not make any claims that this increase signals the US omission or that it drives extinction learning. In contrast, the second voltammetry study by Oleson et al., 2012, observed decreased DA release in the NAc during the CS presentation but no changes in DA release at CS offset (see their Figure 2D) during fear extinction. We are therefore not sure why the reviewer has mentioned this study.

Based on the above considerations we think it’s fair to say that whether DA neurons signal the US omission during fear extinction is incompletely understood. Certainly, it is not accurate to state that it is “well established”, as the reviewer does. We furthermore believe that our study provides several important novel findings that more firmly establish US omission responses of DA neurons during fear extinction and their relationship to extinction learning.

1) We clearly demonstrate that putative DA neurons increase their activity at the time of the US omission during the early trials of fear extinction. Here it is worth emphasizing that for a neuron to be classified as US omission responsive its firing rate had to be significantly above the firing rate both before the CS as well as in the last bin of the CS (see Materials and methods, subsection “Analysis of single-unit data”). Thus our US omission responses cannot simply reflect a delayed response to the CS, which is a possibility in the Badrinarayan et al., 2012 study, as already discussed.

2) We show that at the cellular level, omission responses are exclusively excitatory and are seen only in putative DA but not non-DA neurons. These new results have been added to the revised version of the manuscript in response to reviewer #2.

3) We confirmed the US omission responses of DA neurons using DA neuron-specific fiber photometry measurements.

4) We show that the US omission signal is strongly correlated with extinction learning.

5) We show using bidirectional optogenetic manipulations that this signal drives extinction learning.

In our opinion, these findings add substantially to what has been reported in previous studies and make an important contribution to our understanding of DA neuron signaling during fear extinction.

This concept is consistent with previous studies reporting that the increase in DA firing or DA release caused by an aversive stimulus may reflect the rewarding effects after its termination or the relief caused when it is omitted – a process which is probably very similar to extinction (Brischoux et al., 2009; Budygin et al., 2012; Mirenowicz and Schultz, 1996).

The reviewer raises an interesting point. However, we think it is very much an open question whether the termination of an aversive stimulus is “a process which is probably very similar to extinction”. We are not aware of any study supporting this claim. It should be emphasized that extinction learning begins when an expected aversive stimulus does *not* occur. This is not the same as when an aversive stimulus is *terminated*. In addition, both the Brischoux et al., 2009, and Budygin et al., 2012, studies were performed in *anesthetized* animals. It is therefore not clear whether termination of the aversive stimulus was experienced as rewarding and caused any relief. It is also not clear whether similar responses are seen in awake behaving animals. Although the study by Mirenowicz and Schultz, 1996, mentioned by the reviewer was performed in awake animals they do not report any responses to the termination of an aversive event (air puff) so we are not sure how this study is relevant (in fact, the main conclusion of this paper is that DA neurons do not respond to aversive air puffs). It is also worth emphasizing that none of the studies mentioned examine responses to aversive US *omission* (the reviewer says ‘omitted’ but we assume he means ‘terminated’). Thus, to what extent DA neurons respond to aversive US termination, and whether this is in any way related to the US omission responses we see, is a matter of speculation. Certainly, the results of these studies cannot be viewed as diminishing the novelty of our results.

The finding that DA activity during the omission of an expected aversive outcome is necessary for fear extinction learning seemed to be the most exciting finding of this paper. However, I noticed that the same finding has been published just recently by another group (Luo et al., 2018). In addition, the Johansen group presented evidence that this effect is specifically mediated by NAc-projecting DA neurons – a finding that I felt was missing in the present work. So, it's nice to see replication of this previous work, but the reviewing editors will need to decide how to weigh reproducibility vs. novelty in this case.

The reviewer is indeed correct that Luo et al., 2018, also show that optogenetic inhibition of DA neurons in the VTA impairs fear extinction learning. However, we believe it is not fair/accurate to describe our study simply as a “replication of this previous work”. Our study also presents a number of additional novel findings supporting the role of DA neurons in fear extinction that is not in Luo et al., 2018. Most importantly, we show using both electrophysiology and fiber photometry, that DA neurons are in fact activated by US omission during fear extinction and that this omission response correlates with extinction learning. Such a demonstration is absent in Luo et al., 2018, but is critical for interpreting the effects of optogenetic manipulations in their study and ours. Also, in addition to inhibition of DA neurons, we show that stimulating them at the time of US omission enhances extinction learning, a finding that was not in Luo et al., 2018. We therefore believe that our study and that of Luo et al., 2018, complement each other and together make a strong case for the role of DA neurons in fear extinction learning.

In addition, we do not agree with the reviewer that Luo et al., 2018, provides evidence that the effect on fear extinction learning is specifically mediated by NAc-projecting DA neurons. In fact, they found that inhibition of DA neuron terminals in NAc did *not* have any effect on fear extinction *learning* (see their Figure 3E). Instead, this manipulation resulted in impaired recall of extinction the next day suggesting an effect on the *consolidation* of extinction memory. The same pattern of findings was also observed when the authors inhibited the DA neurons projecting to the medial shell of NAc (Figure 4C). It is well established that learning and consolidation are separate processes (McGaugh, 2000; Johansen et al., 2011). Therefore, which DA projection mediates the effect on fear extinction learning remains an open question. This issue is discussed in the Discussion section of our revised manuscript (seventh paragraph). We agree with the reviewer that identifying the relevant DA projection will be an important goal in future research.

What was surprising to me is that the authors only found a very small number of putative DA neurons (8/30) that exhibit an increase in firing rate to the omission of the aversive US during the early extinction trials (subsection “Dopamine Neurons Signal the Unexpected Omission of the Aversive US during Fear Extinction Learning”, first paragraph and Figure 1). Let's assume the recorded cells are indeed all dopaminergic, then it means that only a very small subpopulation of DA neuron mediates this effect.

In fact, compared to previous studies (Matsumoto and Hikosaka, 2009; Matsumoto et al., 2016; Tian and Uchida, 2015), we observed much stronger DA neuron responses and a larger percentage of cells showing increased activity to the omission of an aversive US (this was already discussed in the original version of the manuscript). That being said, the reviewer is correct that only a subpopulation of DA neurons shows the US omission response and we think this is an important point. It is by now well established that DA neurons are not a homogeneous population but should rather be viewed as a collection of subpopulations that are defined in part by their projection targets and behavioral functions (Roeper, 2013). Recent studies, measuring activity in axonal terminals of DA neurons, have furthermore begun to reveal clear differences in the response properties of these DA neuron subpopulations (see for example Howe and Dombeck, 2016; Menegas et al., 2017; 2018; Parker et al., 2016). It is therefore perhaps not to be expected that the number of neurons showing US omission responses, which also may have a specific projection target, is large. It should also not be very surprising that a subpopulation of neurons can mediate an effect on fear extinction learning; indeed many studies have demonstrated behavioral effects following manipulations of DA neuron subpopulations (Lammel et al., 2012; Matthews et al., 2016; Menegas et al., 2018; Tye et al., 2013).

A limitation of the fiber photometry approach is that there is little information about how uniform the activity is across neurons and changes in net activity may be the dominant pattern of activity of a small subset of neurons. Additionally, the data in Figure 2F suggest that at least some animals also showed an increase in calcium activity during the CS presentation.

We agree with the reviewer that a limitation of fiber photometry is that it reflects the average bulk activity of all DA neurons in the vicinity of the optic fiber. If we understand the reviewer correctly, he also suggests that the US omission response we observe in the fiber photometry signal may be driven by a subset of DA neurons, a possibility which we also agree with. Indeed, our electrophysiology results suggest that only a subset of putative DA neurons show a US omission response. It is also worth pointing out that we only observed excitatory responses to US omission, whereas none of our recorded putative DA neurons displayed inhibitory responses. As a result, a significant excitatory response to US omission can be observed in the average response of the entire putative DA neuron population (now added as Figure 1J in the revised manuscript in response to reviewer #2). Our electrophysiology results thus complement the fiber photometry results by suggesting that a subpopulation of DA neurons consistently respond with excitation to US omission.

The reviewer is also correct to point out that some animals showed an increase in calcium activity during the CS. We have now added a new figure (Figure 4 in the revised manuscript) to show the responses to the CS. We find, in contrast to the uniform increase in DA neuron activity during US omission (Figure 3G), that the responses to the CS varied across animals (Figure 4A), with some animals showing increased and others decreased DA neuron activity (Figure 4AB). Accordingly, the Ca^+2^ signal evoked by the CS was not significantly different from the baseline when we averaged the CS responses of all animals (Figure 4A). These results are also consistent with the CS responses observed in our putative DA neurons which are now shown in Figure 1—figure supplement 4 in the revised manuscript.

It is also possible that the non-responsive cells in Figure 1 are non-dopaminergic (i.e., putative GABAergic or glutamatergic neurons). Thus, another limitation of the present work is that the authors did not unambiguously identify DA neurons. Their conclusions rely upon indirect methods such as spike waveforms and response to rewarding stimuli. These identification methods have been called into question and may not be appropriate for the identification of DA neurons (Margolis et al., 2006, 2010; Ungless and Grace, 2012). Thus, the authors should consider alternative approaches such as an optogenetic tagging method to unambiguously identify DA neurons (and other VTA cell populations) in highly heterogeneous brain regions such as the VTA – they clearly have the technical expertise to do this.

The reviewer raises an important point. It is indeed possible that some of the non-responsive neurons we classified as DAergic might actually be non-DAergic. However, it is important to note that such a misclassification would lead to an underestimation of the proportion of responsive DA neurons but would not change our conclusion that DA neurons signal the omission of an aversive US. More generally, we agree with the reviewer that the classical methods we used to identify DA neurons (i.e. spike waveform analysis) have their limitations. However, it should be emphasized that we used similar identification methods as in the majority of electrophysiology studies that have examined the activity of DA neurons in awake behaving animals. Some of these studies (performed by the labs of Schultz, Hikosaka and Schoenbaum) were the first to identify reward prediction error signaling in dopamine neurons, and similar responses were later observed using opto-tagged DA neurons (by the Uchida Lab). We therefore believe that, although not ideal, the classical waveform analysis methods are still of value. Nonetheless, because of the above mentioned limitations, we sought to confirm our electrophysiology results with DA neuron-specific fiber photometry recordings, a technique that has been used by a number of studies in the DA literature (Cui et al., 2013; Gunaydin et al., 2014; Parker et al., 2016; Matthews et al., 2016; Menegas et al., 2017; 2918; Soares et al., 2016). Importantly, we found similar results using this technique indicating that the single-units that we classified as putative DA neurons were most likely DA neurons.

We also agree with the reviewer that it would have been better to use the opto-tagging method to identify DA neurons. It is not clear to us why the reviewer thinks that we "clearly have the technical expertise to do this". We do not have any publications in which we performed optotagging of DA neurons. This is a method which we have not established sufficiently well for recording from freely behaving animals during fear extinction (it should be emphasized that for each animal there is only one extinction session and thus sampling of different neurons across different sessions is not possible, as it is for many other reward-related behaviors). On the other hand, we have been able to opto-tag DA neurons in head-fixed mice where we record acutely using opto-probes. We have used this approach to address a comment raised by Reviewer #3 (see his point #2, below).

Reviewer #2:[…] 1) One of the important and novel aspects of this study is single neuron recording during fear extinction. Although the authors report the minimum results to support their conclusions, this study can be further strengthened if the authors report a little more detail of the recorded neuron population. The authors first select those that were activated during shock omission and report the average firing pattern of these neurons. This represents only a fraction of all putative dopamine neurons (8 out of 30 putative dopamine neurons). It is important to know how other putative dopamine neurons responded (no response or were some of them inhibited?). Please show the distribution of the responses of all putative dopamine neurons during shock omission. Is the average response of all putative dopamine neurons significantly positive?

We are grateful for the reviewer’s very careful and detailed review of our study. It is correct that we focused our analysis on the putative DA neurons that were activated during the US omission and reported the average firing pattern of only these neurons. We agree with the reviewer that presenting the response profile of other putative DA neurons will further strengthen our study. To address this we have examined whether there were any putative DA neurons in our dataset – which we have increased with the addition of data from 4 new mice – that were inhibited by the US omission. We found that none of the putative DA neurons (0 out of 40 neurons) showed a selective inhibition to the US omission during either early or late extinction. This suggests that the dominant response of putative DA neurons to the omission of the US was excitation. We next looked at the average response of all putative DA neurons, as suggested by the reviewer, and found that it was significantly positive (elevated) at the time of the US omission during early extinction. Notably, this finding is consistent with our results obtained using fiber photometry which measures the average DA neuron population activity. These results are now described in the second paragraph of the subsection “Dopamine Neurons Signal the Unexpected Omission of the Aversive US during Fear Extinction Learning” and in Figure 1J in the revised manuscript. Furthermore, we included a new figure (Figure 2A in revised manuscript) where we show the distribution of responses of all putative DA neurons during US omission and indicate the significantly excited/inhibited neurons. We believe that these new results strengthen the conclusion of our study and we thank the reviewer for his constructive comments and suggestions.

Furthermore, the authors have recorded the activity of unidentified dopamine neurons. Although this is not essential for the authors' conclusion, it would be useful to report how these neurons responded. Are there narrow-spiking neurons that were activated during shock omission? This may have already been done in Figure 1—figure supplement 2: does the red circle represent a putative non-dopaminergic neuron that was activated during shock omission? Do the black circles actually represent putative non-dopaminergic neurons that were not activated during shock omission? Please clarify. Also, it would be useful to describe these results in the main text.

The reviewer brings up a very good point. Local circuit interactions between DA and GABA neurons in the VTA play an important role in reward learning (Cohen at al., 2012; Eshel et al., 2015). Thus, how non-DA neurons in VTA respond during fear extinction is an important and interesting question. Because our main aim was to record the activity of putative DA neurons, we sampled mostly neurons with low baseline firing rate (< 10 Hz) and broad spike waveforms. We therefore had a small number of putative non-DA neurons (narrow-spiking; 18 of 48 neurons) during the extinction session in the original manuscript. To address the issue raised by the reviewer, we have now included data from four new animals and increased the number of non-DA neurons (47 of 90 neurons during tone habituation and 35 of 75 neurons during extinction) substantially in the revised manuscript. Analysis of the activity of putative nonDA neurons revealed that these neurons did not respond to US omission. The proportion of non-DA neurons which showed a significant increase or decrease during the US omission was small (excited: 2.8%, inhibited: 8.5%) and not different than the proportions observed during tone habituation. These results therefore suggest that the US omission was selectively signaled by the putative DA neurons in the VTA. These new results are now described in the third paragraph of the subsection “Dopamine Neurons Signal the Unexpected Omission of the Aversive US during Fear Extinction Learning”. Furthermore, we include a new figure (Figure 2B) in the revised manuscript that shows the distribution of responses of all putative non-DA neurons during the US omission and indicate the significantly excited/inhibited neurons in this figure. In addition, we have clarified and expanded the information shown in Figure 1—figure supplement 2 (Figure 1—figure supplement 3 in the revised manuscript), as suggested by the reviewer. We now show the neurons recorded during tone habituation and extinction on separate figures and on each figure indicate the putative DA and non-DA neurons, as well as the distribution of significantly excited and inhibited neurons. We believe that addition of these new results strengthens our study and we again thank the reviewer for his constructive suggestions.

2) Please report the result of quantification for Figure 2B, 3B, C (TH staining compared with the expression of NpHR or ChR2).

We have now quantified the DA neuron-specific expression of viral constructs by comparing the expression of gCaMP6, NpHR and ChR2 with TH staining. Consistent with previous reports (Lammel et al., 2015), we observed a high level of DA neuron selective expression in DAT-cre mice. The results of the quantification for the fiber photometry and optogenetic experiments are presented in Figure 3—figure supplement 1C (for gCaMP6 and GFP groups), Figure 5—figure supplement 1C (for NpHR and eYFP groups) and Figure 6—figure supplement 1C (for ChR2 and eYFP groups) in the revised manuscript.

3) Figure 2G (the correlation between 'Percent freezing' and 'Average dF/F') is very interesting. However, additional information is needed to interpret this result. First, it is better to use the reduction in freezing rather than the absolute value of freezing, in order to represent learning. Does the correlation hold even if the authors use the difference of freezing between before and during/after extinction? Second, it is a little unclear what is causing the variation in the photometry signal. One possibility, which the authors would like to exclude, is the variation in recording location. To address this, can the authors examine the correlation between freezing (ideally the change in freezing) and reward response? Not obtaining significant correlation in this analysis would strengthen the authors' conclusion.

We thank the reviewer for bringing these points to our attention. Because the freezing levels of all mice were comparable during the beginning of extinction, we used the absolute value of freezing during L-Ext in Figure 2G. However, we agree with the reviewer that it is better to use the change in freezing as a measure of extinction learning. We have therefore performed the correlation again, this time between the change in freezing (from E-Ext to L-Ext) and the average dF/F, and found that these two variables are significantly correlated as well. This new result is now presented in the seventh paragraph of the subsection “Dopamine Neurons Signal the Unexpected Omission of the Aversive US during Fear Extinction Learning” and in Figure 3H of the revised manuscript. The second point that the reviewer raises is a very important issue. It is indeed possible that the variation in the photometry signal can be due to the variation in the recording location rather than reflecting the relationship with extinction learning. To rule this possibility out, we examined the correlation between the change in freezing (E-Ext to L-Ext) and reward responses, as suggested by the reviewer, and did not find a significant relationship between these two variables. This suggests that the variation in the photometry signal we observed during E-Ext was unlikely due to differences in the recording location. We now describe this new result in the aforementioned paragraph and in Figure 3I of the revised manuscript. These results therefore suggest that the correlation between the magnitude of the photometry signal during E-Ext and the level of extinction learning were not due to differences in recording location.

4) Please cite and discuss Luo et al., 2018 and Mayer et al., 2018.

Because Luo et al., 2018, was published after our first submission, this study was not cited in our original manuscript. We have now cited and discussed this study and Mayer et al., 2018, in the Discussion in the revised manuscript. However, it is important to note that Mayer et al., 2018 study investigated relief and safety learning mechanisms using backward conditioning and unpaired conditioning paradigms. In backward conditioning, the order of CS and US presentations is switched such that the CS is delivered at the termination of the aversive US. For studying safety learning, the authors used an explicitly unpaired conditioning protocol in which CS and US presentations were separated in time during the conditioning session. Although these paradigms modeling relief and safety learning might be conceptually similar to fear extinction, procedurally they are quite different. We would like to emphasize that extinction is a form of learning that is initiated when an unexpected event occurs (omission of an expected aversive stimulus) and therefore we believe that the behavioral tasks used in Mayer et al., 2018, are very different than fear extinction.

In addition, we have now also cited and discussed four recent relevant studies (Felsenberg et al., 2018; Groessl et al., 2018; Jo et al., 2018; Menegas et al., 2018) in the subsection “Dopamine Neurons Signal the Unexpected Omission of the Aversive US during Fear Extinction Learning” and in the Discussion.

Reviewer #3:Combining sophisticated electrophysiological, photometric and optogenetic techniques the authors studied the question of how dopaminergic neurons are involved in extinction learning. In their experiments they were able to demonstrate the importance of activity of the midbrain ventral tegmental area (VTA) dopamine system in the initiation and facilitation of cued fear extinction. Activation of dopaminergic neurons is observed when an expected aversive event does not occur, which is a crucial phase in the onset of extinction learning. The study is exciting and in large parts novel. It also provides further information on the neuronal basis of findings of a very recently published study by Luo et al. reporting a role for the VTA-nucleus accumbens dopamine system in extinction learning. Methods are state-of-the art, experiments were carried out with care and the manuscript is straightforward to read. However, there are points that need to be addressed.1) As mentioned above, in a paper published a few weeks ago Luo and co-workers demonstrate that the optogenetic inactivation of VTA-dopamine cells during the US omission period reduced extinction learning in rats. Here, part of this finding is replicated in another species and extended as, for example: a) the optogenetic stimulation of VTA dopaminergic activity facilitated fear extinction in mice (Figure 4); b) (Re)extinction is possible in the paired-NpHR mice during extinction recall when the laser is switched off (Figure 3). This finding may also suggests that silencing of the VTA-dopamine activity did not reconsolidate the fear memory. It is necessary to cite the Luo paper and emphasize the overlap and differences (some of which mentioned above) with the present manuscript.

We are grateful for the reviewer’s careful and detailed reading of our study. Because Luo et al., 2018, was published after submission of our manuscript, this study was not cited in our original manuscript. It is now cited and discussed in detail in the revised version (Discussion).

We thank the reviewer for bringing the issues regarding re-extinction and reconsolidation to our attention and regret that they were not discussed sufficiently in the original manuscript. The reviewer is correct to point out that re-extinction was possible in the paired-NpHR group during extinction recall in the absence of optogenetic manipulation. To address this more directly, we have now compared the extinction learning of paired-NpHR group during extinction recall (day 3) to that of the control groups during extinction learning (day 2). This revealed that there was no significant difference in the extinction rate of these groups. This result suggests that: 1) Paired-NpHR group is capable of undergoing normal extinction learning in the absence of optogenetic manipulation the next day indicating that our manipulation did not have a nonspecific long-term effect, 2) our optogenetic manipulation likely did not have an effect on the strength of the fear memory, as suggested by the reviewer. These results are now presented and discussed in the last paragraph of the subsection “Inhibition of Dopamine Neuron Firing at the Time of the US Omission Impairs Fear Extinction Learning”.

Furthermore, we found that enhancing DA neuron firing at the time of the US omission accelerated extinction learning and enhanced extinction memory. However, the low level of freezing in Paired-ChR2 mice during extinction recall test could also be due to an effect on the fear memory itself. For instance, excitation of DA neurons could have resulted in the erasure of the fear memory by impairing the memory reconsolidation process (Nader, 2015). To rule this possibility out, we did perform a fear renewal test in these mice but didn’t include these results in the original manuscript. These results are now described in the last paragraph of the subsection “Enhancing Dopamine Neuron Firing at the Time of the US Omission Accelerates Fear Extinction Learning” and in Figure 6—figure supplement 3. We show that all three groups of mice exhibited comparable levels of fear renewal suggesting that the low level of freezing in Paired-ChR2 mice during extinction recall cannot be due to impaired memory reconsolidation, but is rather due to enhanced extinction. We believe that these new results strengthen our study and we thank the reviewer for his constructive comments and suggestions.

2) Please provide evidence that optical silencing and stimulation using NpHR and ChR2, respectively, is a valid method to modulate dopaminergic activity in the VTA of mice.

The reviewer raises an important issue. It is indeed essential to provide evidence that optical stimulation of NpHR and ChR2 are valid methods to modulate DA neuron firing in awake mice. To address this issue, we have performed single unit recordings of DA neurons in the VTA of awake DAT-cre mice expressing NpHR or ChR2 during optical stimulation. These results are now presented in Figure 5—figure supplement 2 and Figure 6—figure supplement 2, respectively. These new results confirm that optical activation of NpHR and ChR2 can inhibit and excite DA neuron firing in awake mice. These recordings were performed in awake head-fixed mice acutely using opto-probes.

3) The dopamine system has been suggested to be involved in both the learning and consolidation of fear extinction. The authors nicely demonstrate its role in the initiation of fear extinction learning, but unfortunately do not further discuss this finding in relation to published work. For example, it has been shown that administration of the dopaminergic drug L-DOPA before extinction training was able to initiate fear extinction learning in a mouse model of impaired fear extinction. This discussion needs to be added. Also the consolidation aspect is not addressed sufficiently (one mentioning that "extinction memory was strengthened"). Furthermore, the authors focus on the better-than-expected outcome reward aspect of dopamine in fear extinction in the Discussion. There is increasing evidence that dopamine activity/release may also reflect a prediction error signal that guides extinction learning. Whether and how the present data support this latter idea, is missing and should be added.

We thank the reviewer for bringing to our attention the studies on the role of L-DOPA during fear extinction and regret that they were not cited and discussed in the original manuscript. We have now discussed these findings in the Discussion section. Furthermore, because the focus of our study is on fear extinction learning, we did not discuss the memory consolidation aspect in detail in the original manuscript. However, we agree that this is an important aspect and have now included this discussion and cited the relevant literature on the role of dopamine signaling in mediating the plasticity in the neural circuit underlying fear extinction memories in the Discussion.

The reviewer is correct to point out that we did not emphasize the prediction error aspect of our findings sufficiently in the original manuscript. We agree with the reviewer that the dopamine neuron activity likely reflects a prediction error signal that drives extinction learning. Indeed, our electrophysiology and fiber photometry experiments suggest that the DA signal at the time of the US omission has the properties of a prediction error signal: it was large at the beginning of extinction when the US omission was most unexpected, and decreased significantly during the late trials of extinction when the US omission was no longer unexpected and animals showed significant extinction learning. However, this aspect of our results was only briefly discussed in the original manuscript and we have now emphasized it more in the revised version. We now clearly define prediction error signaling in the Introduction and have also extended our discussion of our results in light of prediction error signaling on in the Introduction and Discussion, as suggested by the reviewer. We believe that discussion of the prediction error and consolidation aspects of our findings in more detail strengthen our study and we thank the reviewer for his constructive comments.

References:

Bouton ME, Woods AM, Todd TP. (2014) Separation of time-based and trial-based accounts of the partial reinforcement extinction effect. Behav Processes. 101, 23-31.

Cui G, Jun SB, Jin X, Pham MD, Vogel SS, Lovinger DM, Costa RM. (2013) Concurrent activation of striatal direct and indirect pathways during action initiation. Nature. 494, 238-42

Grady AK, Bowen KH, Hyde AT, Totsch SK, Knight DC. (2016) Effect of continuous and partial reinforcement on the acquisition and extinction of human conditioned fear. Behav Neurosci. 130, 36-43.

Gunaydin LA, Grosenick L, Finkelstein JC, Kauvar IV, Fenno LE, Adhikari A, Lammel S, Mirzabekov JJ, Airan RD, Zalocusky KA, Tye KM, Anikeeva P, Malenka RC, Deisseroth K. (2014) Natural neural projection dynamics underlying social behavior. Cell. 157, 1535-51.

Howe MW, Dombeck DA. (2016) Rapid signalling in distinct dopaminergic axons during locomotion and reward. Nature. 535, 505-510.

Johansen JP, Cain CK, Ostroff LE, LeDoux JE. (2011) Molecular mechanisms of fear learning and memory. Cell. 147, 509-524.

Lammel S, Lim BK, Ran C, Huang KW, Betley MJ, Tye KM, Deisseroth K, Malenka RC. (2012) Input-specific control of reward and aversion in the ventral tegmental area. Nature. 491, 212-7.

Matthews GA, Nieh EH, Vander Weele CM, Halbert SA, Pradhan RV, Yosafat AS, Glober GF, Izadmehr EM, Thomas RE, Lacy GD, Wildes CP, Ungless MA, Tye KM (2016) Dorsal Raphe Dopamine Neurons Represent the Experience of Social Isolation. Cell 164, 617-631.

McGaugh JL. (2000) Memory--a century of consolidation. Science. 287, 248-251.

Skinner, B. F. (1938). The Behavior of Organisms. New York: Appleton-Century-Crofts.

Tye KM, Mirzabekov JJ, Warden MR, Ferenczi EA, Tsai HC, Finkelstein J, Kim SY, Adhikari A, Thompson KR, Andalman AS, Gunaydin LA, Witten IB, Deisseroth K. (2013) Dopamine neurons modulate neural encoding and expression of depression-related behaviour. Nature. 493, 537-541.